

# On the flow of states under $T\overline{T}$

**Jorrit Kruthoff and Onkar Parrikar**

Department of Physics, Stanford University, Stanford, CA 94305-4060, USA

## Abstract

We study the $T\overline{T}$ deformation of two dimensional quantum field theories from a Hamiltonian point of view, focusing on aspects of the theory in Lorentzian signature. Our starting point is a simple rewriting of the spatial integral of the $T\overline{T}$ operator, which directly implies the deformed energy spectrum of the theory. Using this rewriting, we then derive flow equations for various quantities in the deformed theory, such as energy eigenstates, operators, and correlation functions. On the plane, we find that the deformation merely has the effect of implementing successive canonical/Bogoliubov transformations along the flow. This leads us to define a class of non-local, "dressed" operators (including a dressed stress tensor) which satisfy the same commutation relations as in the undeformed theory. This further implies that on the plane, the deformed theory retains its symmetry algebra, including conformal symmetry, if the original theory is a CFT. On the cylinder the $T\overline{T}$ deformation is much more non-trivial, but even so, correlation functions of certain dressed operators are integral transforms of the original ones. Finally, we propose a tensor network interpretation of our results in the context of AdS/CFT.

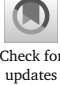

# 1   Introduction

The $T\bar{T}$ deformation of two dimensional quantum field theories provides a concrete set-up to study non-local effects in quantum field theory, in particular those which might arise from coupling the theory to gravity. Due to some remarkable properties of the $T\overline{T}$ operator found by Zamolodchikov [1], it turns out that the spectrum of energy eigenvalues of the deformed theory on the cylinder (i.e., when the spatial slice is a circle) can be solved exactly, given the undeformed spectrum. This spectrum shows some tantalizing properties which are reminiscent of string theory or theories with a UV completion, despite the operator being irrelevant [2–7]. For instance, with a particular sign of the deformation, the spectral density of the theory develops a Hagedorn growth of states. On the other hand, for the opposite sign of the coupling, the energies exactly match with the gravitational quasi-local energies of black holes in $AdS_3$ with a radial cutoff on the asymptotic region [8–10]. This latter feature is particularly interesting because getting rid of the asymptotic region in AdS/CFT would be a very promising starting point in moving towards quantum gravity beyond asymptotically AdS spaces [11].

In the past few years, much effort has gone into understanding various apsects of $T\overline{T}$ deformed quantum field theories, such as the spectrum on the circle and its complexification, sphere and torus partition functions [12–15], the holographic aspect of the $T\overline{T}$ deformation, correlation functions on the Euclidean plane [9, 16, 17] and higher-dimensional generalization [18–20]. Furthermore, a particularly interesting direction is the study of the entanglement structure of states in these (non-local) theories [21–25]. However, it would be fair to say that beyond the deformed energy spectrum and partition functions, many of these aspects are not fully understood. In 0+1 dimensions, i.e., in $T\overline{T}$ deformed quantum mechanics [26–28][1], the deformed spectrum of the theory is all one really needs, as this entirely fixes the correlation functions of the deformed theory. However, in 1+1 dimensions, this is not true – along with the energy eigenvalues, the energy eigenstates of the theory also change under the $T\overline{T}$ deformation, something which is clearly important to keep track of when we study observables such as correlation functions or entanglement entropy. Furthermore, for the holographic sign of the deformation, the flow of eigenstates is intimately tied with the idea of the "surface-state correspondence" proposed in [30, 31] (see also [32]), which was at least in part inspired by the analogy between AdS/CFT and tensor-networks (see, for instance, [33–39]). Our central objective here will be to study the flow of energy eigenstates under the $T\overline{T}$ deformation, and the effect this has on the flow of correlations functions. We hope that our results will also shed some light on other issues such as entanglement entropy, surface-state correspondence/tensor networks in AdS/CFT, etc.

---

[1]See also [29] for an interesting alternative proposal for finite cutoff JT gravity.

## Summary and outline

We will focus primarily on the flow of energy eigenstates, operators and correlation functions in a $T\overline{T}$ deformed quantum field theory in Lorentzian signature. Motivated by the formula for the deformed energy spectrum, plus the results on $T\overline{T}$ deformation in $0+1$ dimensions [26,27], we take as our starting point a definition of the $T\overline{T}$ deformed theory from a Hamiltonian point of view, namely that the Hamiltonian $H_\lambda$ and momentum $P$ of the deformed theory change under the flow as

$$\partial_\lambda H_\lambda = \int dx_1 \, \mathcal{O}^{(\lambda)}_{T\overline{T}}(x_0, x_1), \qquad \partial_\lambda P = 0, \tag{1.1}$$

with $\lambda$ the deformation parameter. The superscript $\lambda$ on the $T\overline{T}$ operator is meant to indicate that the stress tensor is that of the theory at $\lambda$. With this definition, the translation symmetries of the original theory are maintained along the flow. Classically, this definition is equivalent to the definition in terms of flow of the action proposed by Smirnov and Zamolodchikov in [4], but quantum mechanically there could be differences arising from operator ordering related counter-terms. At any rate, we will take the definition (1.1) as our starting point. We will later show that this definition of the $T\overline{T}$ deformation in Lorentzian signature is consistent with the other known results, such as, for instance, the deformed $S$-matrix [2,40].

Given this definition, we begin our analysis in section 2.1 with the following crucial observation: the spatial integral of the $T\overline{T}$ operator can always be written as a sum of two terms

$$\partial_\lambda H_\lambda = i\left[H_\lambda, \mathcal{X}^{(\lambda)}\right] + \mathcal{Y}^{(\lambda)}, \tag{1.2}$$

where explicit expressions for $\mathcal{X}^{(\lambda)}$ and $\mathcal{Y}^{(\lambda)}$ are given in equation (2.10). The first of these terms is clearly a total-in-time derivative; as such it does not change the energy eigenvalues, but merely implements a *canonical transformation* on phase space, or equivalently a *Bogoliubov transformation* on the Hilbert space. A lattice version of this term was also found in [41]. On the other hand, the second term $\mathcal{Y}^{(\lambda)}$ turns out to be a manifestly factorized operator, i.e., a product of two spatial integrals of the stress tensor (see equation (2.10)). This rewriting directly implies the known formula for the deformed energy spectrum of the theory [1,4], and also simplifies the analysis of eigenstates in what follows.

With this observation in hand, we compute various quantities as a function of $\lambda$, both on the plane and cylinder. The most basic ones are the energy eigenstates. Since translation symmetries remain unbroken under the flow, these states $|E(\lambda), k\rangle$ are labelled by the energy and momentum. In case of the spatial topology being a circle, the momentum is quantized in units of the circle length. Due to the $T\overline{T}$ deformation, the energy eigenstates start to mix and we give an explicit expression for the unitary matrix $U$ implementing that mixing in section 2.2. This unitary $U$ depends on the deformed stress tensor and in section 2.3, we rewrite it in terms of a kernel which involves a path integral over a fluctuating "worldsheet", which we dub the *Cauchy string*.

We then turn to the question of correlation functions in section 3. On the plane, we consider correlators of two types of operators – the first type are operators of the original seed theory, but time evolved with the deformed Hamiltonian. We obtain a flow equation for the correlation functions of this class of operators on the plane, which agrees with that of [17] and can be physically interpreted in terms of a "state-dependent diffeomorphism" via the attachement of a stress tensor "Wilson line". The second type of operators are what we call *dressed operators*. The definition of these operators is motivated by the simple rewriting of the spatial integral of the $T\overline{T}$ operator in equation (1.2). In particular, the $\mathcal{Y}^{(\lambda)}$ term drops out on the plane if we restrict attention to finite energy/near-vacuum states, and so the $T\overline{T}$ deformation on the plane acts as a pure canonical transformation in classical terms, or a Bogoliubov transformation

quantum mechanically. With this in mind, the dressed operators are defined as the "canonically transformed" operators, $\widetilde{O} = UOU^{-1}$. These dressed operators have the property that they are causal, i.e. they commute with each other at spacelike separation, and additionally their correlation functions, the structure constants in their commutator algebra etc. are invariant along the flow. However, the dressed operators do not spacelike commute with the operators of the seed theory, i.e., they are non-local with respect to the original seed operator algebra. In particular, we can also construct a (conserved) dressed stress tensor (which we emphasize is different from the *local* stress tensor) such that its correlation functions on the plane, its algebra etc. remain invariant under the flow. A deformed CFT on the plane therefore continues to have a conserved, traceless stress tensor which satisfies the same commutator algebra as in the undeformed CFT, albeit one which is non-local with respect to the seed operators. As an example, we give an explicit expression for the dressed operators in the classical $T\overline{T}$ deformed free, scalar field theory. On the cylinder, the situation with correlation functions is much more complicated and we do not have a complete picture for the flow of operators/correlation functions. Nevertheless, for *dressed* operators, we are able to write the deformed correlation functions as an integral transform of the original correlators, just as in 1d $T\overline{T}$ [27].

In section 4, we briefly discuss how the expected CDD factor in the flat space S-matrix of $T\overline{T}$ deformed theories arises from our analysis. We then give a 2+1 dimensional gravitational viewpoint on the unitary $U$, reminiscent in spirit and form of the gravitational kernels which have appeared previously in [15, 42, 43]. Finally, we also propose a tensor network interpretation of our results in the context of AdS/CFT. We end with some remarks on future directions in section 5.

## 2 Energy eigenstates and their flow

The $T\overline{T}$ deformation is a one-parameter deformation of a quantum field theory, which is often defined from a Lagrangian perspective as a flow of the Lagrangian density of the theory:

$$\partial_\lambda \mathcal{L} = -\mathcal{O}_{T\overline{T}}^{(\lambda)} = -\varepsilon^{ab}\varepsilon^{cd} T_{ac}^{(\lambda)} T_{cd}^{(\lambda)}, \tag{2.1}$$

where $T_{ab}^{(\lambda)}$ is the stress tensor of the theory at the flow parameter $\lambda$. Since the stress tensor can itself be constructed from the Lagrangian density, say by the Noether procedure, this defines a self-contained flow equation for the classical Lagrangian density of the field theory. Quantum mechanically, the common approach is to use the integral of this deformed Lagrangian density as the action inside the Feynman path integral, and this gives a definition for the partition function, generating functional of correlation functions etc. In this paper, we will take a Hamiltonian perspective on the $T\overline{T}$ deformation, i.e. we will define it via a flow of the Hamiltonian of the theory:

$$\partial_\lambda H_\lambda = \int dy_1 \, \varepsilon^{ab}\varepsilon^{cd} T_{ac}^{(\lambda)}(y_0, y_1) T_{bd}^{(\lambda)}(y_0, y_1), \tag{2.2}$$

where we have written this operator on the Cauchy slice at some time $y_0$, with $y_1$ being the spatial coordinate. Note that this was already used in the derivation of the deformed energy spectrum in [1, 4]. Classically, the two definitions are entirely equivalent (see Appendix A). Quantum mechanically, the two may differ by operator-ordering related counterterms. At any rate, we will take equation (2.2) as our starting point, and use it to construct energy eigenstates and correlation functions along the flow.

## 2.1 Rewriting the $T\overline{T}$ operator

We can write the deformation of the Hamiltonian in a somewhat more illuminating way by using the properties of the $T\overline{T}$ operator. We will employ a variant of the Green function method explained in [17] for this purpose. We begin by trivially rewriting the spatial integral of the $T\overline{T}$ operator in equation (2.2) as a double integral at equal times by inserting a spatial delta function:

$$\int dy_1 \mathcal{O}_{T\overline{T}}(y_0, y_1) = \int dy_1 dw_1 \varepsilon^{ab} \varepsilon^{cd} \delta(y_1 - w_1) T_{ac}^{(\lambda)}(y_0, y_1) T_{bd}^{(\lambda)}(y_0, w_1). \tag{2.3}$$

Here the spatial slice can either be compact (in which case we have a circle of length $L$) or non-compact, and correspondingly the Lorentzian spacetime is either a cylinder or a plane. We now rewrite the spatial delta function in terms of the Green function for the spatial derivative, defined as

$$\partial_{y_1} G(y_1 - w_1) = \delta(y_1 - w_1) - \mu, \tag{2.4}$$

where the constant $\mu = 0$ when the spatial slice is non-compact, while for a compact spatial slice we have $\mu = 1/L$ (corresponding to the subtraction of the zero mode of the derivative operator). Explicitly, this Green function is given by

$$G(x) = \frac{1}{2}\text{sgn}(x) \tag{2.5}$$

in the non-compact case (i.e., when $x \in \mathbb{R}$), and

$$G(x) = \sum_{n \in \mathbb{Z}, n \neq 0} \frac{e^{i\frac{2\pi nx}{L}}}{2\pi in} = \frac{1}{2}\text{sgn}(x) - \frac{x}{L} \tag{2.6}$$

in the compact case (i.e., when $x \in [-L/2, L/2]$ with perodic boundary conditions). Replacing the delta function in (2.3) in terms of the Green function, we find

$$\int dy_1 \mathcal{O}_{T\overline{T}}^{(\lambda)}(y_0, y_1) = -\int dy_1 dw_1 \varepsilon^{ab} \left(\partial_{w_1} G(y_1 - w_1) - \mu\right) T_{0a}^{(\lambda)}(y_0, y_1) T_{1b}^{(\lambda)}(y_0, w_1)$$

$$- \int dy_1 dw_1 \varepsilon^{ab} \left(\partial_{y_1} G(y_1 - w_1) + \mu\right) T_{1a}^{(\lambda)}(y_0, y_1) T_{0b}^{(\lambda)}(y_0, w_1). \tag{2.7}$$

Note that we can regulate the Green function $G(y_1 - w_1)$ by requiring it to drop to zero sufficiently fast in the coincident limit $|y_1 - w_1| \ll \epsilon$ for some short distance cutoff $\epsilon$, where the stress tensors are approaching a coincident limit. Alternatively, one could regulate $G$ by truncating the sum in (2.6) at some large $|n| = N_{\text{max}}$. Upon a partial integration,[2] we can rewrite this as

$$\int dy_1 \mathcal{O}_{T\overline{T}}^{(\lambda)}(y_0, y_1) = \int dy_1 dw_1 \varepsilon^{ab} G(y_1 - w_1) T_{0a}^{(\lambda)}(y_0, y_1) \partial_{w_1} T_{1b}^{(\lambda)}(y_0, w_1)$$

$$+ \int dy_1 dw_1 \varepsilon^{ab} G(y_1 - w_1) \partial_{y_1} T_{1a}^{(\lambda)}(y_0, y_1) T_{0b}^{(\lambda)}(y_0, w_1)$$

$$+ \mu \int dy_1 dw_1 \varepsilon^{ab} \varepsilon^{cd} T_{ac}^{(\lambda)}(y_0, y_1) T_{bd}^{(\lambda)}(y_0, w_1). \tag{2.8}$$

---

[2]In the non-compact case, we should keep track of the boundary terms. Instead, here we will work with the cylinder and to get to the plane, take the limit $L \to \infty$ in the end.

Now we can use conservation of the stress tensor, together with the fact that $H$ generates time translations, to finally rewrite this in the following form:

$$\partial_\lambda H_\lambda = \int dy_1 \mathcal{O}_{T\overline{T}}^{(\lambda)}(y_0, y_1) = i\left[H_\lambda, \mathcal{X}^{(\lambda)}(y_0)\right] + \mathcal{Y}^{(\lambda)}(y_0), \tag{2.9}$$

where $\mathcal{X}$ and $\mathcal{Y}$ are given by the following bi-local integrals[3]:

$$\mathcal{X}^{(\lambda)}(y_0) = \int dy_1 dw_1 \varepsilon^{ab} G(y_1 - w_1) T_{0a}^{(\lambda)}(y_0, y_1) T_{0b}^{(\lambda)}(y_0, w_1), \tag{2.10}$$

$$\mathcal{Y}^{(\lambda)}(y_0) = \mu \varepsilon^{ab} \varepsilon^{cd} \mathbf{P}_{ac}(y_0) \mathbf{P}_{bd}(y_0)$$

$$= \mu \left( \{H, \int dy_1 \Theta(y_0, y_1)\} + 2(H^2 - P^2) \right). \tag{2.11}$$

Here we have used the following notation:

$$\mathbf{P}_{ab}(y_0) = \int dy_1 T_{ab}^{(\lambda)}(y_0, y_1), \ \ \Theta = T^{(\lambda)a}{}_a.$$

Equation (2.9) is the main formula we will utilize repeatedly in the following sections.

Note that the first term in (2.9) can be removed by performing a *canonical transformation*. For instance, in the classical theory, this term is of the form $\{H, \mathcal{X}\}_{PB}$, where the subscript PB stands for Poisson brackets. In classical mechanics, such a deformation is generated by a canonical transformation, with the generating function being $\mathcal{X}$.[4] Note however that this generating function $\mathcal{X}$ is not local in space, but rather a bi-local integral. As we will discuss below, the first term in (2.9) thus merely has the effect of "dressing" the fundamental degrees of freedom, while leaving their energies unaffected (see section 3). The $\mathcal{Y}$ term, on the other hand, which is written entirely in terms of spatial integrals of the energy momentum tensor, does change the energy levels of the theory.

## 2.2 Energy eigenvalues and eigenstates

With the simplified form of the spatial integral of the $T\overline{T}$ operator, (2.9), we proceed to study the flow of the energy eigenstates under the $T\overline{T}$ deformation. The flow of energy eigenvalues is already well-understood [1, 4], but we begin by reviewing it briefly. Let us denote the set of deformed energy eigenstates by $\{|n_\lambda\rangle\}$ and the undeformed ones by $\{|n_0\rangle\}$. These states are also simultaneous eigenstates of the momentum operator, with the momentum eigenvalue constant along the flow. We will assume, without loss of too much generality, that for a given initial energy $E_n^{(0)}$ and momentum $k_n$, there is either no degeneracy, or that the degeneracy does not split along the $T\overline{T}$ flow, so we can use non-degenerate perturbation theory. If the degeneracy splits, then we instead need to use degenerate perturbation theory to begin with, but then after that point we can repeat our argument below. In the case of a 2d CFT as the initial theory, there are indeed degeneracies in the energy spectrum, but as was noted in [44], in the situation where these degeneracies arise due to other (commuting) charges, such as the

---

[3]We also note that $\mathcal{X}^{(\lambda)}$ can also be further rewritten as $\mathcal{X}^{(\lambda)} = i\left[H_\lambda, \mathcal{W}^{(\lambda)}\right]$, where

$$\mathcal{W}^{(\lambda)} = \int dx_1 dy_1 G_{\text{Lap.}}(y_1 - w_1) T_{00}^{(\lambda)}(0, y_1) T_{00}^{(\lambda)}(0, w_1),$$

and $G_{\text{Lap.}}$ is the Green function for the Laplacian on the circle/line.

[4]In the language of symplectic geometry, this term arises from a symplectomorphism on phase space, i.e., a diffeomorphism which preserves the symplectic form.

Korteweg-de Vries charges, they do not split along the $T\overline{T}$ flow and so our arguments below apply. With this assumption, recall that under a deformation in the Hamiltonian $\partial_\lambda H_\lambda$, the energies get deformed as

$$\partial_\lambda E_n(\lambda) = \langle n_\lambda | \partial_\lambda H_\lambda | n_\lambda \rangle, \tag{2.12}$$

which from equation (2.9), we can rewrite as

$$\partial_\lambda E_n(\lambda) = i \langle n_\lambda | [H_\lambda, \mathcal{X}^{(\lambda)}] | n_\lambda \rangle + \mu \, \varepsilon^{ab} \varepsilon^{cd} \langle n_\lambda | \mathbf{P}_{ac} \mathbf{P}_{bd} | n_\lambda \rangle. \tag{2.13}$$

The first term above drops out, and the second term, upon using $\mathbf{P}_{00} = H$ and $\mathbf{P}_{01} = P$ gives

$$\partial_\lambda E_n = \frac{2}{L} E_n \int dy_1 \langle n_\lambda | T_{11}(0, y_1) | n_\lambda \rangle - \frac{2}{L} k_n^2, \tag{2.14}$$

where $k_n$ is the momentum eigenvalue of the state $|n\rangle$. Finally, using (see Appendix B)

$$\langle n_\lambda | T_{11}(0, y_1) | n_\lambda \rangle = -\partial_L E_n, \tag{2.15}$$

we arrive at the following differential equation:

$$\partial_\lambda E_n = -2 E_n \partial_L E_n - \frac{2}{L} k_n^2. \tag{2.16}$$

This is the Burger's equation for the flow of energy eigenvalues which was derived in [1, 4]. The solutions to (2.16) are well-known:

$$E_n(\lambda) = \frac{L}{4\lambda} \left( 1 - \sqrt{1 - 8 \frac{\lambda E_n^{(0)}}{L} + 16 \frac{k_n^2 \lambda^2}{L^2}} \right). \tag{2.17}$$

Let us now turn to the flow of energy eigenstates. A standard result from non-degenerate perturbation theory gives

$$\partial_\lambda |n_\lambda\rangle = \sum_{m \neq n} \frac{\langle m_\lambda | \partial_\lambda H_\lambda | n_\lambda \rangle}{E_\lambda^n - E_\lambda^m} |m_\lambda\rangle. \tag{2.18}$$

We simplify this expression replacing the denominator by an integral,

$$\frac{1}{E_\lambda^n - E_\lambda^m + i\epsilon} = -i \int_0^\infty ds \, e^{is(E_\lambda^n - E_\lambda^m + i\epsilon)}, \tag{2.19}$$

with $\epsilon > 0$, which is required to make the integral converge, for any state $|n_\lambda\rangle$ other than the vacuum.[5] Furthermore, using $O(s) = e^{isH} O(0) e^{-isH}$, we find

$$\partial_\lambda |n_\lambda\rangle = -\sum_{m \neq n} i \int_0^\infty ds \, e^{-\epsilon s} |m_\lambda\rangle \langle m_\lambda | \partial_\lambda H_\lambda(-s) | n_\lambda \rangle. \tag{2.20}$$

At this stage, we will need to assume completeness of the $\{|m_\lambda\rangle\}$ basis of states. On the plane, or on the cylinder with $\lambda < 0$ (assuming the ground state energy satisfies $E_0^{(0)} \geq 0$), we expect this to be true. However, on the cylinder with the holographic sign $\lambda > 0$, or in the situation that $\lambda < 0$ but some of the low-lying states in the undeformed spectrum have negative energy, there is a subtlety – in this case some of the energy eigenvalues become complex along the

---

[5]For the vacuum, we could give $s$ a small imaginary part, but this does not work for general excited states.

flow. This also clearly poses a problem for the convergence of the integral in equation (2.19). It is not clear whether one must discard the corresponding states or not, but if one does discard them, then we would need to ensure that $\partial_\lambda H_\lambda$ does not mix between the real and complex energy states. In what follows, we will simply restrict to the plane with either sign of $\lambda$, and the cylinder with $\lambda < 0$ (assuming the ground state energy satisfies $E_0^{(0)} \geq 0$) to avoid the complexification of energies.

So going back to (2.20), using the completeness of the $|m_\lambda\rangle$ basis together with the previous assumption that the degeneracy of states does not change along the flow, we get

$$\partial_\lambda |n_\lambda\rangle = -i \int_0^\infty ds\, e^{-\epsilon s} \partial_\lambda H_\lambda(-s)|n_\lambda\rangle + i \int_0^\infty ds\, e^{-\epsilon s} \langle n_\lambda|\partial_\lambda H_\lambda(-s)|n_\lambda\rangle |n_\lambda\rangle. \tag{2.21}$$

The above differential equation can be solved by making the following ansatz for the state $|n\rangle_\lambda$:

$$|n\rangle_\lambda = e^{i\theta_n(\lambda)} U(\lambda)|n\rangle_0, \tag{2.22}$$

where $U$ is a unitary operator, and we have pulled out an eigenstate-dependent phase from it. In terms of this ansatz, equation (2.21) then translates to

$$\partial_\lambda U = -i \int_{-\infty}^0 ds\, e^{\epsilon s} e^{isH_\lambda} \partial_\lambda H_\lambda e^{-isH_\lambda} U, \quad \partial_\lambda \theta_n = \frac{1}{\epsilon} \partial_\lambda E_n, \tag{2.23}$$

with formal solution given by,

$$U = \mathcal{P} \exp\left(-i \int_0^\lambda d\lambda' \int_{-\infty}^0 ds\, e^{\epsilon s} \partial_{\lambda'} H_{\lambda'}(s)\right), \quad \theta_n(\lambda) = \frac{1}{\epsilon}(E_n(\lambda) - E_n(0)). \tag{2.24}$$

Finally, using $\partial_\lambda H = \int d\theta\, \mathcal{O}_{T\overline{T}}$, the operator $U$ in (2.24) can be rewritten as

$$U = \mathcal{P} \exp\left(-i \int_0^\lambda d\lambda' \int_{M_-} e^{\epsilon s} \mathcal{O}_{T\overline{T}}\right), \tag{2.25}$$

where $M_- = \mathbb{R}_- \times \Sigma$, with $\Sigma = \mathbb{R}$ or $S^1$. Note that if we try to naively take (2.24) to be true even in the cases where the energy spectrum complexifies, then the $e^{i\theta_n}$ factor would either diverge or decay. The form of $U$ we have obtained in (2.25) is rather formal, but we can get some further intuition in two ways. Firstly, by performing some manipulations using equation (2.9), the above $U$ can be re-written in terms of a kernel, which can be interpreted as the Cauchy slice becoming "dynamical", with the dynamics controlled by a string worldsheet action. We will present this in the next subsection. Secondly, one can also use the random metric approach of [45] where one interprets the $T\overline{T}$ deformation as coupling the seed theory to a random metric. This leads to an effective, three dimensional gravitational kernel for the unitary $U$ (similar in spirit to [5, 15, 42]). We will defer this 3d approach to section 4.

## 2.3 A kernel for $U$

Going back to equation (2.9), the unitary operator $U$ can now be expressed in terms of the bi-local operators $\mathcal{X}$ and $\mathcal{Y}$ as

$$U = \mathcal{P} \exp\left[-i \int_0^\lambda d\lambda' \left(\mathcal{X}^{(\lambda')}(0) + \mu \int_{-\infty}^0 ds\, e^{\epsilon s} \varepsilon^{ab} \varepsilon^{cd} \mathbf{P}_{ac}^{(\lambda')}(s) \mathbf{P}_{bd}^{(\lambda')}(s)\right)\right]. \tag{2.26}$$

Note that the $\mathcal{X}$ term entirely localizes on the $s = 0$ spatial slice.[6] The second term proportional to $\mu$ is more complicated and involves operators at finite time, but at least on the plane, this term drops out. At any rate, this expression for the unitary $U$ makes it fairly easy to write a flow equation for correlation functions in the $T\overline{T}$ flowed CFT, as we will show in section 3 below. Note that equation (2.26) is strikingly reminiscent of tensor networks [33–39] and the surface-state correspondence [30,31] in the context of AdS/CFT, at least on the plane ($\mu = 0$); we will return to this point later.

We can also rewrite this expression in terms of a path-integral kernel involving a "string worldsheet" as follows (see figure 1). We first break up the path-ordered exponential into infinitesimal exponentials:

$$U = \lim_{\delta\lambda \to 0} \prod_{k=0}^{N} U_k, \;\; U_k = \exp\left[-i\delta\lambda \int_{M_-} e^{\epsilon s} O_{T\overline{T}}(\lambda_k = k\delta\lambda)\right], \qquad (2.27)$$

where $N = \lambda/\delta\lambda$. Now using equation (2.26), each of these infinitesimal unitaries can be written as

$$U_k = \exp\left[-i\delta\lambda\,\mathcal{X}^{(\lambda_k)}(0) - i\delta\lambda\mu\left(\left\{H_{\lambda^k}, \int_{M_-} \Theta^{(\lambda_k)}\right\} - \frac{2}{\epsilon}(P^2 - H_\lambda^2)\right)\right], \qquad (2.28)$$

where we have rewritten $T_{11}$ in terms of the trace of the stress tensor $\Theta$. Next, we rewrite this as

$$U_k = \int [D\xi_k(\sigma)DQ_kD\phi_k]\exp\left[i\delta\lambda S[\xi_k,Q_k,\phi_k] - i\delta\lambda \oint d\sigma\,\xi_k^a(\sigma)T_{0a}^{(\lambda_k)}(0,\sigma)\right.$$
$$\left. - i\delta\lambda\left(Q_k^0 H + Q_k^1 P\right) - i\delta\lambda\phi_k \int_{M_-} \Theta^{(\lambda_k)}\right], \qquad (2.29)$$

where

$$S[\xi_k,Q_k,\phi_k] = \frac{1}{4}\oint d\sigma\,\epsilon_{ab}\xi_k^a(\sigma)\partial_\sigma\xi_k^b(\sigma) - \frac{\epsilon}{8\mu}(Q_k^1)^2 - \frac{1}{2\mu\epsilon}\phi_k^2 + \frac{1}{2\mu}\phi_k Q_k^0, \qquad (2.30)$$

For each $k$th infinitesimal piece we have introduced a vector valued Hubbard-Stratanovich (HS) field $\xi_k^a(\sigma)$ which only depends on the spatial coordinate, a vector valued HS field $Q^a$ and a scalar HS field $\phi$ both of which are spacetime independent. We can combine $Q^a$ and $\xi^a(\sigma)$ into one field, with $Q^a$ being the zero mode and $\xi^a$ being the remaining non-zero modes, whose spatial integral vanishes. In fact, it is more convenient to define a field $X^a(\lambda,\sigma)$ such that

$$\partial_\lambda X^a(\lambda,\sigma) = Q^a(\lambda) + \xi^a(\lambda,\sigma). \qquad (2.31)$$

Now sending $\delta\lambda \to 0$, we can rewrite the full unitary $U$ as a path integral over the fields $X^a$ and $\phi$:

$$U = \int \frac{[DXD\phi]}{\mathcal{N}}e^{i(S+S_{\text{reg}})}\mathcal{P}\exp\left[-i\int_0^\lambda d\lambda'\left(\oint d\sigma\,\partial_{\lambda'}X^a\,T_{0a}^{(\lambda')}(\sigma) + \phi(\lambda')\int_{M_-}\Theta^{(\lambda')}\right)\right] \quad (2.32)$$

where the action is given by

$$S[X,\phi] = \frac{1}{4}\int_0^\lambda d\lambda'\left(\oint d\sigma\,\varepsilon_{ab}\partial_{\lambda'}X^a\partial_\sigma\partial_{\lambda'}X^b + 2\phi(\lambda')\oint \partial_{\lambda'}X^0\right), \qquad (2.33)$$

---

[6]We have taken the $\epsilon \to 0$ limit in the $\mathcal{X}$ term and dropped an $O(\epsilon)$ term resulting from integration by parts.

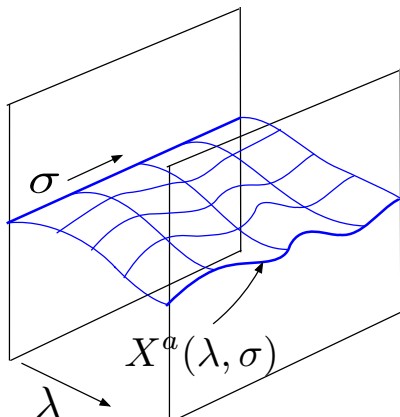

Figure 1: We can interpret the unitary $U$ as making the Cauchy slice a dynamical surface parametrised by $X^a(\lambda, \sigma)$.

and the term $S_{\text{reg}}$ regularizes the zero mode integrals:

$$S_{\text{reg}} = -\frac{1}{2} \int_0^\lambda d\lambda' \left( \frac{\epsilon\mu}{2} \oint d\sigma \oint d\sigma' \partial_{\lambda'} X^1(\sigma) \partial_{\lambda'} X^1(\sigma') + \frac{1}{\mu\epsilon} \phi(\lambda')^2 \right). \tag{2.34}$$

We can interpret the $X^a$ field in terms of an effective "Cauchy string" (see figure 1). The coordinate $\sigma$ is an intrinsic coordinate along the string, and $\lambda$ is an emergent Euclidean "time" direction, parametrizing the $T\overline{T}$ flow. $X^a(\lambda, \sigma)$ is then a map of the Cauchy string worldsheet to the target space, which is either $\mathbb{R}^2$ or $\mathbb{R} \times S^1$. Therefore, we may interpret the unitary $U$ as making the Cauchy slice in the CFT a dynamical object, in a manner of speaking. From the tensor network perspective mentioned above, we seem to have a superposition of tensor networks, at least on the plane. The interpretation of the $\phi$ field is not clear to us at this point, but it roughly seems to be a dilaton-like field implementing a rescaling of the cylinder.

## 3 Flow of operators and correlation functions

In the previous section, we have shown how the energy eigenstates change under the flow triggered by the $T\overline{T}$ operator. In particular, we found an explicit form of a unitary operator $U$ that rotates these states amongst each other. Next, we would like to know how correlation functions change under the flow (pertubatively in $\lambda$ such correlator have been computed, for instance, see [9][7] for a perturbative approach). This requires knowing how operators flow. There are several different approaches one could consider for the flow of operators/correlation functions. Here, we consider two type of operators:

*(i)* The first type of operators, which we will call *undeformed operators*, are those obtained from time evolution of the operators of the undeformed theory. More precisely, we consider some constant time Cauchy slice, say at $t = 0$, and consider the undeformed operators $O(0, x)$ of the seed theory on this Cauchy slice. Operators at a time separation away from the Cauchy slice are of course defined in the usual way via

$$O^{(\lambda)}(t, x) = e^{itH_\lambda} O(0, x) e^{-itH_\lambda}, \tag{3.1}$$

and since the Hamiltonian of the theory is changing along the flow, these finite time operators will also change, but only via their dependence on $H_\lambda$. The one exception to this is the stress

---

[7]See also [46, 47]

tensor – since the Hamiltonian is $H_\lambda = \int dx\, T_{00}^{(\lambda)}(0,x)$, we are forced to let $T_{\mu\nu}^{(\lambda)}(0,x)$ change explicitly along the flow. At least classically, this explicit flow of $T_{\mu\nu}^{(\lambda)}(0,x)$ can be obtained via Noether's procedure from the flow of the Lagrangian density of the theory.

*(ii)* The second type of operators we will consider are what we will call *dressed operators*, where we explicitly flow the operators on the initial time slice. This flow is motivated by the observation that the $T\overline{T}$ deformation on the plane can be removed by a canonical/Bogoliubov transformation. The operators at finite time are then again defined in the usual way via time evolution.

## 3.1 On the plane

For simplicity of presentation, we first consider the case of the theory on the plane and then on the cylinder. Again, we mention here that our discussion below applies only to finite energy states. If the energy of the state under consideration is not finite, but has a finite energy density, the $T\overline{T}$ deformation can change the energy density and the flow is morally similar to the one on the cylinder.

**Undeformed operators**

We will first consider correlation functions of the undeformed operators defined above. Let us consider the following correlation function:

$$C(\{t_i, x_i\}) = \langle n_\lambda | O^{(\lambda)}(t_1, x_1) \cdots O^{(\lambda)}(t_n, x_p) | n_\lambda \rangle, \tag{3.2}$$

where $|n_\lambda\rangle$ is an energy eigenstate with energy $E_n$. We can derive a flow equation for this correlation function as follows: we first insert complete sets of energy eigenstates between the operators:

$$C = \sum_{n_1, \cdots, n_{p-1}} e^{it_1(E_n - E_{n_1}) + \cdots + it_p(E_{n_{p-1}} - E_n)} \langle n_\lambda | O(0, x_1) | n_{1,\lambda} \rangle \cdots \langle n_{p-1,\lambda} | O(0, x_p) | n_\lambda \rangle. \tag{3.3}$$

Now we can use the fact that the energy eigenvalues on the plane are $\lambda$-independent, and so also are the operators $O(0, x_i)$ on the initial time slice, as per our choice. Therefore, the only $\lambda$-dependence in the correlation function comes from the energy eigenstates, which satisfy the following flow equation:

$$
\begin{aligned}
\partial_\lambda |n_\lambda\rangle &= -i \int_{-\infty}^{0} ds\, e^{\epsilon s} \partial_\lambda H_\lambda(s) |n_\lambda\rangle \\
&= -i \int_{-\infty}^{0} ds\, e^{\epsilon s} e^{isH_\lambda} i\big[H_\lambda, \mathcal{X}^{(\lambda)}\big] e^{-isH_\lambda} |n_\lambda\rangle \\
&= -i \int_{-\infty}^{0} ds\, e^{\epsilon s} \partial_s \big(e^{isH_\lambda} \mathcal{X}^{(\lambda)} e^{-isH_\lambda}\big) |n_\lambda\rangle = -i \mathcal{X}^{(\lambda)} |n_\lambda\rangle.
\end{aligned}
\tag{3.4}
$$

Note that we have dropped the $\mathcal{Y}$ term above, assuming that it is suppressed in the $L \to \infty$ limit at finite energy. Since we are primarily interested in vacuum correlation functions, we expect this to be a good assumption. Therefore, taking a $\lambda$ derivative of the correlation function (3.3) gives

$$\partial_\lambda C = i \sum_{i=1}^{p} \langle n_\lambda | O^{(\lambda)}(t_1, x_1) \cdots \big[\mathcal{X}^{(\lambda)}(t_i), O^{(\lambda)}(t_i, x_i)\big] \cdots O^{(\lambda)}(t_p, x_p) | n_\lambda \rangle, \tag{3.5}$$

where we have defined

$$\mathcal{X}^{(\lambda)}(t) = \frac{1}{2} \int dy \int dw \, G(y-w) \varepsilon^{ab} T_{0a}^{(\lambda)}(t,y) T_{0b}^{(\lambda)}(t,w). \tag{3.6}$$

Note that on general grounds the commutator can be simplified,

$$\left[ \mathcal{X}^{(\lambda)}(t_i), O^{(\lambda)}(t_i,x_i) \right] = \int dy \, G(y-x_i) \varepsilon^{ab} T_{0a}(t_i,y) \partial_b^{(x_i)} O^{(\lambda)}(t_i,x_i) + \cdots, \tag{3.7}$$

where $\cdots$ denotes a theory-dependent, local operator, which, if we like, we can absorb via a local redefinition of the operators $O^{(\lambda)}$. Equations (3.5) and (3.7) agree with the flow equation for correlation functions derived recently by Cardy in [17] using Euclidean path integral methods, up to the local operator re-definitions mentioned above. As suggested in [17], we can figuratively think of the effect of the $T\overline{T}$ deformation on correlation functions as implementing a "state-dependent diffeomorphism" via the attachment of a stress tensor "Wilson line" to the operators. Despite the non-locality of this "Wilson line", we emphasize that that since the operators on the initial time slice are those of the undeformed theory, their equal-time commutators at separate points will continue to vanish inside correlation functions. Furthermore, since the deformation preserves Lorentz invariance on the plane, commutators of more general spacelike separated operators will also continue to vanish. The non-local Wilson line attachment in the flow equation obscures the above causal properties of these correlation functions, nevertheless we expect their analytic structure to still be controlled by causality.

Equation (3.7) is a bit formal, because we need to address the *UV* divergencies which appear in the limit when the two operators becomes co-incident. As discussed previously, we can regulate these divergences by introducing a short distance cutoff $\epsilon$ in the Green function $G$, such that it drops to zero when $|y-x| \ll \epsilon$. Fortunately, these UV divergences were addressed in the analysis of Cardy in [17], where it was shown that the RHS of (3.7) only has a logarithmic divergence in $\epsilon$ :

$$\int dy \, G(y-x_i) \varepsilon^{ab} T_{0a}(t_i,y) \partial_b^{(x_i)} O^{(\lambda)}(t_i,x_i) = -\log(|\epsilon|) \nabla_i^2 O^{(\lambda)}(t_i,x_i) + \text{finite}. \tag{3.8}$$

Crucially, note that the divergence in the flow equation is proportional to a local operator, and thus corresponds to a cutoff-dependent, local redefinition of the operator at every step along the flow. In other words, we should locally redefine the operators $O^{(\lambda)}$ at every step along the flow in order to cancel the above divergence.

Equation (3.5) gets slightly modified if one of the operators in the correlation function is the stress tensor. In this case we need to account for the explicit change in the stress tensor on the initial time slice along the flow. As mentioned previously, this explicit change in the stress tensor can be obtained, at least classically, from Noether's procedure:

$$\partial_\lambda T_{ij}^{(\lambda)}(\phi,\dot{\phi}) = \partial_i \phi \frac{\delta}{\delta \partial^j \phi} \left( \varepsilon^{ab} \varepsilon^{cd} T_{ac}^{(\lambda)} T_{bd}^{(\lambda)} \right) - \eta_{ij} \varepsilon^{ab} \varepsilon^{cd} T_{ac}^{(\lambda)} T_{bd}^{(\lambda)} + \cdots, \tag{3.9}$$

where $\phi$ denotes the elementary fields in the action and $\cdots$ denote potential improvement terms which may be required to make the stress tensor symmetric. An additional subtlety is that the above stress tensor is written in terms of $\phi$ and its time derivatives, but the operator written in terms of the canonical variables $(\phi,\pi)$ will have an additional contribution of the form $(\partial_\lambda \dot{\phi}) \frac{\delta}{\delta \pi} T_{ij}^{(\lambda)}$ coming from the change in the relation between $\pi$ and $\dot{\phi}$. All these contributions to correlation functions appear to be theory dependent.

**Dressed operators**

Now we come to the second type of operators of interest to us, which we will call *dressed* operators and will only be considered in detail on the plane for reasons that will be clear momentarily. To motivate the definition of these dressed operators, we go back to equation (2.9), which implies that the spatial integral of the $T\overline{T}$ operator on the plane (i.e., at $\mu = 0$) is given by

$$\partial_\lambda H_\lambda = i\left[H_\lambda, \mathcal{X}^{(\lambda)}\right], \quad \mathcal{X}^{(\lambda)} = \frac{1}{2}\int dx_1 \int dy_1\, G(x_1 - y_1)\varepsilon^{ab} T_{0a}^{(\lambda)}(0, x_1) T_{0b}^{(\lambda)}(0, y_1), \quad (3.10)$$

where we again emphasize that we have dropped the $\mathcal{Y}$ term above in the $L \to \infty$ limit, assuming that we are working at finite energy. It is helpful to first look at the classical analog of equation (3.10), which is

$$\partial_\lambda H_\lambda = \left\{H_\lambda, \mathcal{X}^{(\lambda)}\right\}_{PB}, \quad (3.11)$$

where the subscript $PB$ stands for Poisson brackets. It is clear that such a deformation of the Hamiltonian can be removed by a canonical transformation, generated by $\mathcal{X}$. In more detail, say that the theory at $\lambda$ is naturally written in terms of some canonical degrees of freedom $(\phi_I^\lambda, \pi_J^\lambda)$ satisfying

$$\left\{\phi_I^\lambda, \pi_J^\lambda\right\}_{PB} = \delta_{IJ}, \quad (3.12)$$

where the $I, J$ are meant to be generalized indices, including the spatial dependence of these fields. Then deforming the Hamiltonian, as in (3.11), is equivalent to keeping the Hamiltonian function unchanged but deforming the phase space coordinates as

$$\partial_\lambda \phi_\lambda^I = -\left\{\mathcal{X}^{(\lambda)}, \phi_\lambda^I\right\}_{PB}, \quad \partial_\lambda \pi_\lambda^I = -\left\{\mathcal{X}^{(\lambda)}, \pi_\lambda^I\right\}_{PB}. \quad (3.13)$$

This flow of phase space coordinates is a canonical transformation/symplectic diffeomorphism, i.e. it preserves the Poisson brackets in (3.12). Thus, classically the $T\overline{T}$ deformation on the plane and at finite energy merely has the effect of implementing a $\lambda$-dependent canonical transformation along the flow. Quantum mechanically, we can replace the Poisson brackets above with commutators, and then it becomes evident that the flow simply implements a unitary rotation on phase space which preserves the canonical commutation relations, or in other words, a *Bogoluibov transformation*.

This motivates us to define the *dressed operators* $\widetilde{O}$ on the initial time slice via the following flow equation:

$$\partial_\lambda \widetilde{O}^{(\lambda)} = -i\left[\mathcal{X}^{(\lambda)}, \widetilde{O}^{(\lambda)}\right]. \quad (3.14)$$

This flow is rather formal, since we have not discussed *UV* divergencies, but we will see that inside correlation function these operators do make sense. We can recast (3.14) in the form

$$D_\lambda \widetilde{O}^{(\lambda)} \equiv \partial_\lambda \widetilde{O}^{(\lambda)} + i\left[\mathcal{X}^{(\lambda)}, \widetilde{O}^{(\lambda)}\right] = 0,$$

where we may think of the derivative $D_\lambda$ defined above as a *covariant derivative*. From this point of view, the dressed operators are covariantly constant along the flow. The flow equation has a simple solution:

$$\widetilde{O}^{(\lambda)}(0, x) = U\, O(0, x)\, U^{-1}, \quad U = \mathcal{P}\, e^{-i\int_0^\lambda d\lambda'\, \mathcal{X}^{(\lambda')}}, \quad (3.15)$$

where the unitary $U$ is the same operator we considered in the previous section. From equation (3.10), it follows that dressed operators at time $t$ are also related to the operators of the seed theory similarly, i.e.,

$$\widetilde{O}^{(\lambda)}(t,x) = U\,O(t,x)\,U^{-1}, \quad O(t,x) = e^{itH_0}O(0,x)e^{-itH_0}. \tag{3.16}$$

The *dressed* operators are thus non-local, since $U$ is non-local.

Note that the dressed operators satisfy the same commutation relations as the seed operators; in particular, dressed operators commute with other dressed operators at spacelike separation. It should perhaps be emphasized that the "dressing" $\mathcal{X}^{(\lambda)}$ is non-local, and so a dressed operator will *not* necessarily commute with an undeformed operator at a spacelike separated point. Nevertheless, the dressed operators do respect causality in that they commute with other dressed operators at spacelike separation, notwithstanding the non-locality of the dressing. Further, since energy eigenvalues on the plane do not flow and eigenstates flow by the action of the same unitary $U$, correlation functions of dressed operators are $\lambda$-independent:

$$\partial_\lambda \widetilde{C}(\{t_i, x_i\}) = \partial_\lambda \langle n_\lambda | \widetilde{O}^{(\lambda)}(t_1, x_1) \cdots \widetilde{O}^{(\lambda)}(t_p, x_p) | n_\lambda \rangle = 0. \tag{3.17}$$

Thus, the dressed operators are a canonical choice of operators along the flow in terms of which the theory appears completely undeformed.

It is important to stress here that while the classical version of the flow equation (3.14) (with Poisson brackets in place of commutators) is perfectly well defined, the quantum version is somewhat formal, since it suffers from the coincident divergences discussed around equation (3.8). Indeed, as discussed there, the right hand side of (3.14) has a local, logarithmic divergence. Thus, if we compute the correlation functions of these dressed operators in the original, undeformed vacuum state, then these will be divergent. However, the vacuum state of the theory also flows and the correlation functions in the flowed vacuum are indeed finite and unchanged. We emphasize that the sole reason this construction works is that the classical version of the deformation is a canonical transformation, i.e., a redefinition of the phase space coordinates. This is no longer the case on the cylinder, or even for finite energy density states on the plane, and in those situations, there is no natural way to define such dressed operators. [8]

Nevertheless, having said that, it remains curious why these operators (on the plane) only make sense inside correlation functions and at present we do not have a full understanding of it. Furthermore, this also touches upon the question whether two canonically related classical theories give the same quantum theory or not. It would be interesting to study this aspect of the $T\bar{T}$ deformation more.

We can also define a *dressed* stress tensor $\widetilde{T}_{ij}$ in the same way as any other operator:

$$D_\lambda \widetilde{T}^{(\lambda)}_{\mu\nu} = 0. \tag{3.19}$$

This is *not* the same as the original stress tensor of the theory which was discussed in the previous section (see equation (3.9)). The dressed stress tensor is not local with respect to the undeformed operators, however it is local (i.e., microcausal) with respect to dressed operators. Furthermore, it is conserved and its spatial integrals give the expected energy-momentum

---

[8]Alternatively, one can subtract off the $UV$ divergence along the flow as proposed in [17]. The flow of the operator would then be defined as

$$D_\lambda \widehat{O}^{(\lambda)}_{ij}(x) = -\log|\mu\varepsilon|\nabla^2_x \widehat{O}^{(\lambda)}_{ij}(x), \tag{3.18}$$

with $\mu$ a renormalisation scale. Correlation functions of these operators in the undeformed state are finite, but are divergent in the deformed state as mentioned in the main text. Our perspective is therefore a little different than [17] as we define correlation functions of dressed operators in the deformed states. This is a natural way to define them, because of the $T\bar{T}$ deformation is a canonical/Bogoliubov transformation on the plane.

charges. To show conservation, it is enough to show that if the dressed stress tensor is conserved at $\lambda$, then the dressed stress tensor at $\lambda + d\lambda$ will also be conserved. To this end, consider the conservation equation and take a $\lambda$ derivative, replacing spacetime derivatives with commutators:

$$\partial_\lambda \left( \partial^\mu \widetilde{T}_{\mu\nu}^{(\lambda)} \right) = \partial_\lambda \left( -i \left[ H_\lambda, \widetilde{T}_{0\nu}^{(\lambda)}(x) \right] + i \left[ P, \widetilde{T}_{1\nu}^{(\lambda)}(x) \right] \right). \tag{3.20}$$

Bringing the $\lambda$ derivative inside the commutators and using (2.9), we can write this as

$$\partial_\lambda \left( \partial^\mu \widetilde{T}_{\mu\nu}^{(\lambda)} \right) = \left[ \left[ H_\lambda, \mathcal{X}^{(\lambda)} \right], \widetilde{T}_{0\nu}^{(\lambda)}(x) \right] - i \left[ H_\lambda, \partial_\lambda \widetilde{T}_{0\nu}^{(\lambda)}(x) \right] + i \left[ P, \partial_\lambda \widetilde{T}_{1\nu}^{(\lambda)}(x) \right]. \tag{3.21}$$

The double commutator can be simplified using the Jacobi identity and after a little algebra, using conservation of $\widetilde{T}_{\mu\nu}^{(\lambda)}(x)$, we find

$$\partial_\lambda \left( \partial^\mu \widetilde{T}_{\mu\nu}^{(\lambda)} \right) = \partial^\mu \left( D_\lambda \widetilde{T}_{\mu\nu}^{(\lambda)}(x) \right) - i \left[ \left[ P, \mathcal{X}^{(\lambda)} \right], \widetilde{T}_{1\nu}^{(\lambda)} \right] = 0, \tag{3.22}$$

where we have used the fact that the dressed stress tensor is covariantly constant, by definition, and that $\left[ P, \mathcal{X}^{(\lambda)} \right] = 0$. Finally, since the dressed stress tensor matches onto the conserved stress tensor of the seed theory at $\lambda = 0$, we conclude that it is conserved everywhere along the flow. Next, the dressed energy and momentum operators obtained from the dressed stress tensor:

$$\widetilde{H}_\lambda = \int dx_1 \widetilde{T}_{00}^{(\lambda)}(0, x_1), \quad \widetilde{P}_\lambda = \int dx_1 \widetilde{T}_{01}^{(\lambda)}(0, x_1), \tag{3.23}$$

satisfy the following flow equations

$$\partial_\lambda \widetilde{H}_\lambda = \int dx_1 \partial_\lambda \widetilde{T}_{00}^{(\lambda)}(0, x_1) = -i \int dx_1 \left[ \mathcal{X}^{(\lambda)}, \widetilde{T}_{00}^{(\lambda)}(0, x_1) \right] = -i \left[ \mathcal{X}^{(\lambda)}, \widetilde{H}_\lambda \right], \tag{3.24}$$

$$\partial_\lambda \widetilde{P}_\lambda = \int dx_1 \partial_\lambda \widetilde{T}_{01}^{(\lambda)}(0, x_1) = -i \int dx_1 \left[ \mathcal{X}^{(\lambda)}, \widetilde{T}_{01}^{(\lambda)}(0, x_1) \right] = -i \left[ \mathcal{X}^{(\lambda)}, \widetilde{P}_\lambda \right]. \tag{3.25}$$

These first order flow equations for $\widetilde{H}_\lambda$ and $\widetilde{P}_\lambda$ are the same as their untilded counterparts and since they have the same $\lambda = 0$ limit, the tilded and untilded charges are the same. Note however that the dressed and undressed stress-tensor are still different and the equality of the charges merely states that they are related through improvement terms, albleit non-local ones. Also notice that even though the flow of the tilded stress tensor is formally divergent, the flow of the charges is not, because the divergence comes with a Laplacian. Its spatial part drops out because of the spatial integral and the temporal part gives a time derivative of the commutators $[\widetilde{H}_\lambda, \widetilde{H}_\lambda]$ and $[\widetilde{P}_\lambda, \widetilde{H}_\lambda]$, which thus also vanishes. Thus, the energy and momentum operators obtained from the dressed stress tensor are the correct energy and momentum operators of the deformed theory.

Finally, if the seed theory is a conformal field theory, then the stress tensor of the seed theory is expected to satisfy an algebra of the form:

$$\left[ T_{\mu\nu}(x), T_{\rho\sigma}(x') \right] = f_{\mu\nu\rho\sigma}^{\alpha\beta}(x - x') T_{\alpha\beta}(x) + \gamma_{\mu\nu\rho\sigma}(x - x'), \tag{3.26}$$

where $f_{\mu\nu\rho\sigma}^{\alpha\beta}$ are the structure constants and $\gamma_{\mu\nu\rho\sigma}$ the central terms. Either by using the flow equation, or by using equation (3.15), it is straightforward to show that the dressed stress tensor $\widetilde{T}_{ij}^{(\lambda)}$ also satisfies the same algebra, with $\lambda$-independent structure constants and central terms. In particular this has the interesting consequence that the dressed stress tensor

behaves like the stress tensor of the seed conformal field theory, with the central charge equal to that of the seed theory, i.e. the Schwinger terms are equivalent. To be a bit more explicit, let us consider the seed theory to be a 2d CFT. This theory has, amongst the usual Lorentz and special conformal currents, a dilatation current $j_\mu^D = T_{\mu\nu}x^\nu$. In the deformed theory this current is simply,

$$\widetilde{j}_\mu^D = \widetilde{T}_{\mu\nu}^{(\lambda)}x^\nu, \tag{3.27}$$

and the charge $\widetilde{D}$ is the spatial integral of $\widetilde{j}_0^D$. Note, however, that this current is *non*-local Commuting this charge (at equal time) with a dressed operator $\widetilde{O}^{(\lambda)}(x)$ it will have the same eigenvalue, i.e. conformal dimension $\Delta$, as in the undeformed theory. This can also be seen from the fact that the correlators of dressed operators do not flow. An interesting question is whether the global conformal group lifts to a full Virasoro symmetry. In these *non-local* CFTs this is far from obvious and we will discuss this further in section 5. Again here we mention that although these results are true classically, in the quantum theory there are UV divergencies that need to be dealt with. However, there are two pieces of evidence why this might not be a such a big issue. [9] First, inside correlation functions of the deformed state these cancel and we end up with well-defined Ward identities. Second, the flow of the conserved charge is again finite by the same argument as given above about the conserved charges associated to translations in space and time.

Finally, one might wonder whether it is possible to define a new flow where at every step one adds to the Hamiltonian the $T\overline{T}$ operator made out of the dressed stress tensor. It is easy to check that in this case, the generating functional $\widetilde{\mathcal{X}}$ is $\lambda$-independent, because

$$\partial_\lambda \widetilde{\mathcal{X}}^{(\lambda)} = -i\left[\widetilde{\mathcal{X}}^{(\lambda)}, \widetilde{\mathcal{X}}^{(\lambda)}\right] = 0, \tag{3.28}$$

and so such a deformation would be equivalent to the "one-shot" deformation where we turn on $\lambda$ times the $T\overline{T}$ operator of the seed theory.

**Example: Classical, free scalar field**

Let us apply the discussion above to a simple example. Let the seed theory be a free, massless scalar field theory on the plane:

$$\mathcal{L}^{(0)} = \frac{1}{2}\left(\dot{\phi}^2 - \phi'^2\right), \tag{3.29}$$

where $\dot{\phi} = \partial_t\phi$ and $\phi' = \partial_x\phi$. Classically, the deformed action corresponding to this seed theory was calculated in [2], and is given by the Nambu-Goto action:

$$\mathcal{L}^{(\lambda)} = \frac{1}{4\lambda}\left(-1 + \sqrt{1 + 4\lambda\left(\dot{\phi}^2 - \phi'^2\right)}\right). \tag{3.30}$$

The canonical momentum conjugate to $\phi$ is given by

$$\pi = \frac{\delta\mathcal{L}^{(\lambda)}}{\delta\dot{\phi}} = \frac{\dot{\phi}}{\sqrt{1 + 4\lambda\left(\dot{\phi}^2 - \phi'^2\right)}}, \tag{3.31}$$

from which we can easily obtain $\dot{\phi}$ as a function of $\pi$. In the Hamiltonian perspective, the canonical variables $(\phi, \pi)$ on an initial time slice (say, $t = 0$) are to be regarded as $\lambda$-independent

---

[9]It would be interesting and of great value to understand this better as there could be subtleties at the quantum level.

field variables, while $\dot{\phi}^{(\lambda)}(\phi, \pi)$ is $\lambda$-dependent. We will often suppress the explicit $\lambda$-dependence of $\dot{\phi}$, but the reader should bear this in mind. The Hamiltonian is given by

$$H_\lambda = \int dx\, h_\lambda(x), \quad h_\lambda = \frac{1}{4\lambda}\left(1 - \sqrt{(1-4\lambda\pi^2)(1-4\lambda\phi'^2)}\right). \tag{3.32}$$

Note that the Hamiltonian density at finite $\lambda$ can be rewritten in terms of that of the seed theory as

$$h_\lambda(x) = \frac{1}{4\lambda}\left(1 - \sqrt{1 + 16\lambda^2 p_0^2 - 8\lambda h_0}\right), \tag{3.33}$$

where $h_0 = \frac{1}{2}(\pi^2 + \phi'^2)$ and $p_0 = \pi\phi'$ are the energy and momentum density of the seed theory. The (canonical) stress tensor can be obtained using Noether's procedure:

$$T_{ij}^{(\lambda)} = -\partial_i \phi \frac{\delta \mathcal{L}^{(\lambda)}}{\delta \partial^j \phi} + \eta_{ij}\mathcal{L}^{(\lambda)}. \tag{3.34}$$

Applying this to the action (3.30), we find

$$T_{00}^{(\lambda)} = \frac{1}{4\lambda}\left(1 - \sqrt{(1-4\lambda\pi^2)(1-4\lambda\phi'^2)}\right) = h_\lambda, \quad T_{01}^{(\lambda)} = \pi\phi' = p_0, \tag{3.35}$$

and

$$T_{11}^{(\lambda)} = \frac{1}{4\lambda}\left\{-1 + 4\lambda\phi'^2\sqrt{\frac{1-4\lambda\pi^2}{1-4\lambda\phi'^2}} + \sqrt{\frac{1-4\lambda\phi'^2}{1-4\lambda\pi^2}}\right\}, \tag{3.36}$$

where we observe that the momentum density $p_\lambda(x)$ at finite $\lambda$ is actually $\lambda$-independent at $t = 0$. One can readily check that this stress tensor satisfies the flow equation $\partial_\lambda T_{00} = \varepsilon^{ab}\varepsilon^{cd}T_{ac}T_{bd}$. From here, we can compute the generator of the canonical transformation:

$$\mathcal{X}^{(\lambda)} = \int dx\, dy\, \text{sgn}(x - y)h_\lambda(x)p_0(y). \tag{3.37}$$

If we have some observable $\mathcal{O}(\phi, \pi)$ in the seed theory, then the corresponding dressed observable $\widetilde{\mathcal{O}}^{(\lambda)}(\phi, \pi)$ can be obtained by solving the following flow equation

$$\partial_\lambda \widetilde{\mathcal{O}}^{(\lambda)}(\phi, \pi) = -\left\{\mathcal{X}^{(\lambda)}, \widetilde{\mathcal{O}}^{(\lambda)}(\phi, \pi)\right\}_{PB}. \tag{3.38}$$

This equation may look complicated because of the $\lambda$-dependence in $\mathcal{X}^{(\lambda)}$, but a closer look at equations (3.37) and (3.33) reveals that we can transform this into a $\lambda$-independent flow by defining the new variables (assuming, for convenience, $\lambda > 0$):

$$x = \sqrt{\lambda}\,\widehat{x}, \quad \phi(x) = \widehat{\phi}(\widehat{x}), \quad \pi(x) = \frac{1}{\sqrt{\lambda}}\widehat{\pi}(\widehat{x}). \tag{3.39}$$

Note that this change of phase space coordinates is also a canonical transformation, i.e., it preserves the Poisson brackets. Thus, we can rewrite equation (3.38) in these new variables as

$$\lambda\partial_\lambda \widetilde{\mathcal{O}}^{(\lambda)}(\widehat{\phi}, \widehat{\pi}) = -\left\{\widehat{\mathcal{X}}, \widetilde{\mathcal{O}}^{(\lambda)}(\widehat{\phi}, \widehat{\pi})\right\}_{PB}, \tag{3.40}$$

where we have defined the new $\lambda$-independent generator $\widehat{\mathcal{X}}$ as

$$\widehat{\mathcal{X}} = \mathcal{D} + \mathcal{K}, \tag{3.41}$$

where we have defined

$$\mathcal{D} = \int dx \, x \widehat{T}_{01}(x), \quad \mathcal{K} = \frac{1}{2} \int dx \, dy \, \text{sgn}(x-y) \varepsilon^{ab} \widehat{T}_{0a}(x) \widehat{T}_{0b}(y), \tag{3.42}$$

and the hatted stress tensor is defined in terms of $\widehat{\phi}$ and $\widehat{\pi}$:

$$\widehat{T}_{00} = \frac{1}{4}\left(1 - \sqrt{(1 - 4\widehat{\pi}^2)(1 - 4\widehat{\phi}'^2)}\right), \quad \widehat{T}_{01} = \widehat{\pi}\widehat{\phi}', \tag{3.43}$$

and does not depdent explicitly on $\lambda$ anymore. Thus, in these dimensionless variables, the flow equation for the dressed observables becomes $\lambda$-independent. We can also rewrite equation (3.40) in terms of a $\lambda$-independent vector field $\mathcal{V}$ on phase space:

$$\lambda \partial_\lambda \widetilde{\mathcal{O}}^{(\lambda)}(\widehat{\phi}, \widehat{\pi}) = -\int dx \left[ \mathcal{V}^{\widehat{\pi}} \frac{\delta}{\delta \widehat{\pi}(x)} + \mathcal{V}^{\widehat{\phi}} \frac{\delta}{\delta \widehat{\phi}(x)} \right] \widetilde{\mathcal{O}}^{(\lambda)}(\widehat{\phi}, \widehat{\pi}), \tag{3.44}$$

where $\mathcal{V}^{\widehat{\pi}} = \frac{\delta \widehat{\mathcal{X}}}{\delta \widehat{\phi}}$ and $\mathcal{V}^{\widehat{\phi}} = -\frac{\delta \widehat{\mathcal{X}}}{\delta \widehat{\pi}}$. The vector field $\mathcal{V}$, which, in the language of symplectic geometry is the Hamiltonian vector field dual to the generating function $\widehat{\mathcal{X}}$, entirely encodes the flow of the dressed observables. At any rate, the key point is that $\mathcal{V}$ is $\lambda$-independent, and so we can formally integrate this flow:

$$\widetilde{\mathcal{O}}^{(\lambda)}(\widehat{\phi}, \widehat{\pi}) = e^{\log(\frac{\lambda_0}{\lambda}) \int dx \left[ \mathcal{V}^{\widehat{\pi}}(x) \frac{\delta}{\delta \widehat{\pi}(x)} + \mathcal{V}^{\widehat{\phi}}(x) \frac{\delta}{\delta \widehat{\phi}(x)} \right]} \widetilde{\mathcal{O}}^{(\lambda_0)}(\widehat{\phi}, \widehat{\pi}). \tag{3.45}$$

This gives an explicit, albeit formal, construction of the classically dressed observables in this theory. Above, we saw that the flow equation for the dressed observables could be expressed in terms of a $\lambda$-independent flow. Although we have only shown this in the special example of the classical, free scalar field, we expect this phenomenon to be generally true of all $T\overline{T}$ deformed CFTs on the plane. If so, the path-ordering in the unitary $U$ can be removed very generally for CFTs on the plane, by repeating the same argument above. Furthermore, equation (3.40) seems to fit nicely within the circle of ideas involving tensor networks (especially the MERA) and the surface state correspondence in AdS/CFT, if we interpret the operator $\mathcal{K}$ above as a "disentangler". We will return to this point in the next section.

## 3.2 On the cylinder

In contrast with the plane, we do not have a complete picture of how operators/correlation functions behave on the cylinder. We present some preliminary results below.

**Undeformed operators**

We can define the undeformed operators on the cylinder in the same way as we did for the plane – we take the operators on an initial time slice to be those of the seed theory (except for the stress tensor), and then operators at a time separation away are defined by time evolution with the deformed Hamiltonian. Even so, correlation functions on the cylinder are much more complicated because both energy eigenvalues and eigenstates change along the flow. For simplicity, let us consider a two-point function of two scalar operators in the vacuum:

$$G_\lambda(t, x) = \langle 0_\lambda | O^{(\lambda)}(t, x) O^{(\lambda)}(0, 0) | 0_\lambda \rangle. \tag{3.46}$$

By inserting a complete set of energy eigenstates of the deformed theory, this correlator can be rewritten as

$$G_\lambda(t, x) = \sum_n |\langle 0_\lambda | O(0, 0) | n_\lambda \rangle|^2 e^{-it\Delta E_n(\lambda)} e^{-ik_n x}, \tag{3.47}$$

with $\Delta E_n = (E_n - E_0)$ is the energy relative to the ground state energy in the deformed theory. Analogously to the Euclidean computation of the finite temperature partition function [5,48], we rewrite the exponential factors using an integral transform,

$$e^{-it\Delta E_n(\lambda)-ik_n x} = \int d^2 x' K_\lambda(t,x;t',x') e^{-it'\Delta E_n(0)-ik_n x'}. \tag{3.48}$$

We can obtain the kernel $K_\lambda$ by a suitable Wick rotation of the contour of integration from the Euclidean formula in [5,48]:

$$K_\lambda(t,x;t',x') = -\frac{tL}{8\pi\lambda}\frac{1}{t'^2}\exp\left(\frac{L}{8i\lambda t'}\left(-(t-t')^2+(x-x')^2\right)\right). \tag{3.49}$$

The integration region in (3.48) for $x'$ is the full real line, whereas for $t'$ it lies on the positive real axis. With this kernel, we can write the deformed correlator as an integral transform of the undeformed one,

$$G_\lambda(t,x) = \int d^2 x\, K_\lambda(t,x;t',x')\widehat{G}(t',x'), \tag{3.50}$$

with

$$\widehat{G}(t',x') = \langle 0_0|e^{it'H_0}U^{-1}O(0,x')Ue^{-it'H_0}U^{-1}O(0,0)U|0_0\rangle. \tag{3.51}$$

**Dressed operators**

Given that the deformation on the cylinder is not a pure canonical transformation, it is not immediately clear how we should define dressed operators. We will provisionally[10] define them as a generalization of (3.15) in the plane case:

$$\widetilde{O}^{(\lambda)}(0,x) = UO(0,x)U^{-1},\quad U = \mathcal{P}\,e^{-i\int_0^\lambda d\lambda'\left(\mathcal{X}^{(\lambda')}+\mu\int_{-\infty}^0 ds\,e^{\varepsilon s}\mathcal{Y}(s)\right)}, \tag{3.52}$$

or in terms of a flow equation, we have

$$\partial_\lambda\widetilde{O}^{(\lambda)}(0,x) = -i\left[\mathcal{X}^{(\lambda)},\widetilde{O}^{(\lambda)}(0,x)\right]-i\mu\int_{-\infty}^0 ds\,e^{\varepsilon s}\left[\mathcal{Y}(s),\widetilde{O}^{(\lambda)}(0,x)\right], \tag{3.53}$$

where recall that $\mathcal{Y} = \varepsilon^{ac}\varepsilon^{bd}\mathbf{P}_{ab}(s)\mathbf{P}_{cd}(s)$, with $\mathbf{P}_{ab}(s) = \oint dx\,T^{(\lambda)}_{ab}(s,x)$. Operators at finite time can be obtained by time evolution with the deformed Hamiltonian. The dressing in the cylinder case is substantially more complicated because of the presence of the term proportional to $\mu$. Correlation functions of these operators are, nevertheless, simpler; for instance the two-point function is given by

$$\widetilde{G}_\lambda(t,x) = \int d^2 x\, K_\lambda(t,x;t',x')G_0(t',x'),\quad G_0(t',x') = \langle 0_0|O(t,x')O(0,0)|0_0\rangle, \tag{3.54}$$

where $G_0$ is the two-point function in the original seed theory and $K_\lambda$ given in (3.49). Given the difficulties in computing the unitary matrix $U$ and the flow of the stress tensor needed to compute the deformed matrix elements, the deformed correlator of dressed operators is remarkably simple and does not suffer from these difficulties, which partly justifies their definition. Unlike the plane case, however, correlation functions of dressed operators do flow on the cylinder – they are merely smeared versions of the seed correlation functions, with the

---

[10]It would be worthwhile to see whether this definition makes sense in the full quantum theory

smearing function $K_\lambda$. This can be thought of as the two dimensional version of the prescription put forward in [26, 27] for computing deformed correlation functions in quantum mechanics. A slightly different point of view can be obtained through a differential equation for the deformed correlator, again inspired from the one for the torus partition function [12, 13]. The change in the energy levels then follows from the differential equation. It is straighforward to check that the appropriate differential operator acting on $\widetilde{G}_\lambda$ is

$$\frac{iL}{2}\partial_\lambda \widetilde{G}_\lambda(t,x) = \left[ t(\partial_x^2 - \partial_t^2 - E_0^2) - 2\lambda\left(\partial_t - \frac{1}{t}\right)\partial_\lambda \right]\widetilde{G}_\lambda(t,x). \tag{3.55}$$

This point of view has the advantange, that we do not need to worry about the existence of a kernel and analytic continuation. From here we can actually also see the smearing. For instance, consider small $\lambda$, then the only term on the LHS that is going to contribute is the Laplacian on 2d Minkowski space. The differential equation then looks like a diffusion equation with $\lambda$ playing the role of an additional fictitious time, and the diffusion constant $D \sim t/L$.

From this differential equation we can actually learn some more. Consider for instance chiral correlators, say $\widetilde{G}_\lambda(x_+)$, then the differential equation for that correlator becomes,

$$\frac{iL}{2}\partial_\lambda \widetilde{G}_\lambda(x_+) = \left[ -\lambda\partial_+\partial_\lambda + \frac{4\lambda}{x_+ - x_-}\partial_\lambda \right]\widetilde{G}_\lambda(x_+), \tag{3.56}$$

whose solution is the undeformed chiral correlator $G_0(x_+)$, since the other solution depends on $x_-$. We thus see that not only the energy eigenvalues of states with $E = k$ do not flow, also chiral correlators are independent of $\lambda$.

Thusfar we have only considered correlators of scalar operators. For the stress tensor we expect the flow of correlation functions to be much more complicated. To calculate, for instance, the entanglement entropy of a region on the circle using twist operators such correlation functions and their flows would be required. We leave the study of these computations to future work and discuss them briefly in the discussion section.

We would also like to define a dressed stress tensor. However, naively defining the dressed stress tensor in the same way as in (3.53) is not enough; we want to ensure that the dressed stress tensor is conserved and that its spatial integrals reproduce the energy and momentum operators. One can check that a naive definition of the dressed stress tensor following (3.53) violates the conservation condition. However, we can deduce the appropriate flow for the stress tensor by studying the conservation equation. Following the same steps leading to equation (3.22) in the plane case, we get on the cylinder:

$$\partial_\lambda\left(\partial^\mu \widetilde{T}_{\mu\nu}^{(\lambda)}\right) = \partial^\mu\left(D_\lambda \widetilde{T}_{\mu\nu}^{(\lambda)}(x)\right) - i\left[\mathcal{Y}, \widetilde{T}_{1\nu}^{(\lambda)}\right], \tag{3.57}$$

where $D_\lambda = \partial_\lambda + i[\mathcal{X}^{(\lambda)}, \cdot]$ is the same covariant derivative defined previously, and recall that $\mathcal{Y} = \mu\varepsilon^{ab}\varepsilon^{cd}\mathbf{P}_{ac}\mathbf{P}_{bd}$. Therefore, conservation of the dressed stress tensor implies:

$$\partial^\mu D_\lambda \widetilde{T}_{\mu\nu}^{(\lambda)}(x) = i[\mathcal{Y}, \widetilde{T}_{0\nu}^{(\lambda)}(x)]. \tag{3.58}$$

From here, it is possible to extract the flow equation for the deformed stress tensor. The final expressions are a bit complicated, so we will present them in Appendix C. Note, however, that this flow equation for the dressed stress tensor is different from that of the other operators we guessed in equation (3.53), and this implies that the commutation relations of the dressed stress tensor with itself and with the other dressed operators will not be preserved along the flow. In particular, we have not checked whether the dressed stress tensor satisfies microcausality (i.e., whether it commutes at spacelike separation with the other dressed operators). It would be nice to understand the causality structure of these dressed operators, or to see if one can define a fully causal set of dressed operators; we leave this to future work.

# 4 Further developments

## 4.1 $S$-matrix

So far we have discussed (arguably) the most important players in a field theory: the operators, spectrum and correlation functions. By knowing how these objects change under the $T\overline{T}$ flow, we know, in principle, everything there is to know about the deformed theory. In this section, we will consider the $S$-matrix on the plane. This quantity has been discussed extensively [49–51] and here we give yet another derivation from our perspective.

Let us start with the $T\overline{T}$ deformed theory on the plane at some value of the coupling $\lambda$. We wish to ask how the S-matrix of the theory changes when we flow from $\lambda \to \lambda + \delta\lambda$. To set up a scattering process, we need to define in and out states at the asymptotic past and future. In the undeformed theory, such states where constructed using insertions of particle creation and annihilation operators at the past and future null infinities. As a result of the $T\overline{T}$ deformation, these operators will now get dressed in the same way as was discussed in section 3, i.e., $a_{p^i} \to U a_{p_i} U^{-1}$. At any rate, the momenta of these particles will be taken as an input for the $S$-matrix computation. We then deform $\lambda \to \lambda + \delta\lambda$, and ask how the S-matrix changes under this deformation. This is given by

$$S_{\lambda+\delta\lambda} = \lim_{t\to\infty} {}_{\text{out}}\langle p'_1, \cdots, p'_m | \mathcal{T} e^{-i\delta\lambda \int_{-t}^{t} dt' \partial_\lambda H(t')} | p_1, \cdots p_n \rangle_{\text{in}}. \tag{4.1}$$

Using the fact that $\partial_\lambda H = i\left[H, \mathcal{X}^{(\lambda)}\right]$, i.e. $\partial_\lambda H$ is a total time-derivative, we learn that the deformation only gives rise to boundary terms at asymptotic infinity:

$$S_{\lambda+\delta\lambda} = \lim_{t\to\infty} {}_{\text{out}}\langle p'_1, \cdots, p'_m | e^{-i\delta\lambda \mathcal{X}^{(\lambda)}(t)} e^{i\delta\lambda \mathcal{X}^{(\lambda)}(-t)} | p_1, \cdots p_n \rangle_{\text{in}}. \tag{4.2}$$

We can conveniently rewrite the contribution at past asymptotic infinity by introducing a Hubbard-Stratanovich field:

$$e^{i\delta\lambda \mathcal{X}^{(\lambda)}(-t)} | p_1, \cdots p_m \rangle_{\text{in}} = \int [D\xi] e^{-2i\delta\lambda \int du \left(\varepsilon_{ab} \xi^a(u) \partial_u \xi^b(u) + \xi^a T_{0a}^{(\lambda)}(-t,u)\right)} | p_1, \cdots p_n \rangle_{\text{in}}, \tag{4.3}$$

where $u$ is a coordinate along the asymptotic spatial slice which approaches past infinity in the limit $t \to \infty$. There is a similar term coming for future infinity as well. If we now take the action of $T_{0a}$ on the in state to be given by $T_{0a}(u) = \sum_{i=1}^{n} p_a^i \delta(u_i - u)$ to represent the $n$-particle in state, and similarly account for the term from future infinity, we precisely land on the gravitational dressing proposed in [40] and therefore the $S$-matrix,

$$S_{\lambda+\delta\lambda}(\{p^i\}) = e^{-i\frac{\delta\lambda}{2} \sum_{i<j} \varepsilon_{ab} p_a^i p_b^j} S_\lambda(\{p^i\}), \tag{4.4}$$

where we have collectively denoted all the in and out momenta by $\{p^i\}$ in this last formula. Since the momenta are $\lambda$-independent, we can trivially integrate this w.r.t $\lambda$ to get the finite $\lambda$ result, which is precisely the CDD factor which has appeared in the previous literature. This derivation of the $S$-matrix is slightly different from what is done in some of the other works using thermodynamic Bethe ansatz. There, one assumes that the $S$-matrix changes by a CDD factor, i.e. the phase in (4.4) and then shows that this is consistent with the spectrum coming from the $T\overline{T}$ deformation. Here we went the other way and took the flow of the Hamiltonian as a starting point.

Finally, let us remark that the dressing of the in and out states through the operator $U$, is analogous to the dressing of asymptotic states by clouds of soft photons in QED as pioneered by Faddeev and Kulish [52]. Just as in QED, the full Hamiltonian in $T\overline{T}$ deformed theories does not just become the free one in the asymptotic past and future and one is forced to define dressed asymptotic states.

## 4.2 $3d$ **gravity interpretation of** $U$

One other straightforward way of simplifying $U$ is by employing a Hubbard-Stratanovich transformation with a symmetric two-tensor field $h_{ab} \sim \partial_\lambda \gamma_{ab}$ directly on (2.25), employing the ideas of [45] (see also [53]). We do so by following similar steps as in (2.3), which we will not flesh out again here. It turns out the unitary $U$ can be rewritten as

$$U = \frac{1}{\mathcal{N}} \int \mathcal{D}\gamma e^{iS[\gamma]} \mathcal{P} \exp\left(-\frac{i}{2}\int_0^\lambda d\lambda' \int_{M_-} d^2x \, e^{\epsilon s/2}\sqrt{\gamma}\partial_\lambda\gamma^{ab}T_{ab}^{(\lambda')}\right), \qquad (4.5)$$

where

$$S[\gamma;\lambda] = \frac{1}{16}\int d^3x\sqrt{\gamma}\left(\partial_\lambda\gamma_{ab}\partial_\lambda\gamma^{ab} + (\gamma^{ab}\partial_\lambda\gamma_{ab})^2\right), \qquad (4.6)$$

and the field $\partial_\lambda\gamma^{ab}(\lambda,x)$ is a $\lambda$-dependent symmetric two-tensor which plays the role of the Hubbard-Stratanovich field inserted at each infinitesimal step along the flow. This is the finite $\lambda$ version of the Gaussian average over $h$ proposed in [45]. The deformation parameter $\lambda$ has thus geometrized in a third direction, alongside the space-time coordinates. This already hints towards a holographic interpretation, to which we shall now come. That this is not the usual AdS/CFT correspondence, should be clear because so far the initial theory can be any 2d theory. This was already noted in [15] and the bulk geometry for which $\lambda$ is a coordinate was referred to as the *fake* bulk.

Furthermore, notice that this path integral has three boundaries. This is not only due to the finite range of the $\lambda$ integral running from 0 to some finite $\lambda$, but also because $M_-$ has a boundary at $t = 0$.

In fact, it is not difficult to show that (4.6) is equivalent to a gauge-fixed path integral of Einstein gravity in AdS$_3$. To see this, let us consider incorporating the metric $\gamma_{ab}$ in a 3d metric in Fefferman-Graham gauge as follows,

$$ds^2 = g_{\mu\nu}dx^\mu dx^\nu = \frac{d\lambda^2}{4\lambda^2} + \frac{2\pi G_N}{\lambda}\gamma_{ab}(\lambda,\{x^k\})dx^a dx^b\,. \qquad (4.7)$$

Here $g_{ab} = 2\pi G_N \gamma_{ab}/\lambda$ the metric on constant $\lambda$ surfaces. Using this foliation, we can write the various derivatives in (4.6) in terms of extrinsic curvature, which by using Gauss-Codazzi can be written as scalar curvatures and boundary terms. Some detials are given in appendix D. The result is

$$S = \frac{1}{16\pi G_N}\left[\int d^3x\sqrt{g}(R+2) + 2\int_\Sigma \sqrt{g_0}d^2x\,(K-1) - 2\int_{\hat\Sigma} d^2x\sqrt{\hat g_0}\hat K - \int d^3x\sqrt{\gamma}R^{(\gamma)}\right]. \qquad (4.8)$$

This is the standard Euclidean Einstein-Hilbert action in AdS$_3$ with both timelike ($\Sigma$) and spacelike ($\hat\Sigma$) boundaries. Here $g_0$ is the induced metric on the boundary and the hatted quantities refer to those on the spacelike boundary at $t = 0$.

## 4.3 Surface-state correspondence and tensor networks

The $T\overline{T}$ deformation is particularly exciting in holographic theories, because with the positive sign of $\lambda$ (in our conventions), it can be interpreted as the theory dual to a bulk quantum theory of gravity in AdS space with a radial cutoff. Thus, the $T\overline{T}$ flow corresponds to the holographic renormalization group flow [54–56] in these theories [8, 19]. An interesting circle of ideas in this context is the tensor network interpretation of the holographic duality, which suggests

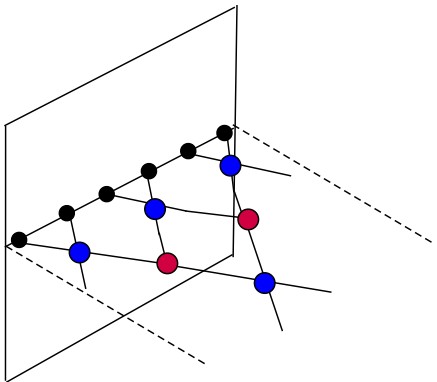

Figure 2: A cartoon of a tensor network. The black dots are initial Hilbert space degrees of freedom, say spins. The blue dots denote tensors which act on the spins as disentanglers while the red dots act as isometries. The emergent geometry of the network is reminiscent of the bulk geometry in AdS/CFT.

that the bulk Cauchy slice should be thought of as a tensor network. A tensor network, in particular the MERA [33, 34, 57–63], is a variational ansatz for the wavefunctions of states in a CFT, which makes key use of the entanglement structure of these states from a position-space renormalization group perspective. In particular, the wavefunction is built as a quantum circuit, with successive layers of local operations called "disentanglers" and "isometries". The rough idea is that starting from the UV state, at every layer of the circuit the disentanglers remove entanglement in the wavefunction at a given length scale, while the isometries coarse-grain and redefine the effective degrees of freedom relevant at the lower energy scale, and this process is repeated scale by scale, until in the end we are left with a completely product state with no entanglement. This "emergent geometry" associated with the tensor network is clearly reminiscent of the bulk geometry in AdS/CFT (see figure 2), as has been discussed in [33–39].

Motivated by this, it was conjectured in [30, 31][11] that every radial slice of a bulk Cauchy surface in AdS/CFT corresponds to a state which is related to the asymptotic CFT state by a unitary transformation. The explicit form of this unitary was not given in these references, but our analysis with the $T\overline{T}$ deformation now gives us some handle on this unitary. For instance, on the plane we have:

$$U = \mathcal{P} \exp\left\{-\frac{i}{2}\int_0^\lambda d\lambda' \int dx_1 dy_1 \text{sgn}(x_1 - y_1)\varepsilon^{ab} T_{0a}^{(\lambda')}(0, x_1) T_{0b}^{(\lambda')}(0, y_1)\right\}. \qquad (4.9)$$

This unitary clearly has the structure of a tensor network, albeit in the continuum, with the bi-local operator in $T_{0a}^{(\lambda)}$ constituting the elementary operations at scale $\lambda$. In fact for a CFT on the plane, when written in terms of dimensionless degrees of freedom as in the example of the free scalar field in section 3 (see equations (3.41)), the unitary organizes in terms of $\lambda$-independent elementary "gates", consisting of a dilatation generator $\mathcal{D}$ plus an operator which we labelled $\mathcal{K}$ in (3.42). This seems to fit in nicely with the tensor network picture, if we regard $\mathcal{D}$ as being an isometry and $\mathcal{K}$ as being the disentangler. We do not have a sharp argument for why we should think of $\mathcal{K}$ as a disentangler, but it is a bi-local operator, and it seems reasonable to think that it adds/removes entanglement between the two points upon which it acts, similar to the Gao-Jafferis-Wall deformation [66]. On the other hand, the unitary also admits another, perhaps more natural, interpretation – the path-integral kernel which was constructed for $U$ in section 2.3 implies that it is a *superposition* of local tensor networks, with the tensors/gates at

---

[11]See also [64, 65] for previous discussion on the surface-state correspondence related to $T\overline{T}$ .

each step consisting of the stress tensor $\xi^a(\lambda, x)T^{(\lambda)}_{0a}(0, x)$ in this interpretation. Such tensor networks/circuits have been previously considered in [67] (see also [68]), but the difference here is that the network generated by the $T\overline{T}$ flow has coefficients $\xi^a$ which must be integrated over with the action derived in section 2.3. It would be nice to understand these points further, as this may lead us to a very explicit realization of the AdS/tensor network correspondence. It would also be interesting to see if there is a connection to the path integral interpretation of $T\overline{T}$ put forward in [20].

# 5 Discussion

The $T\overline{T}$ deformation of two-dimensional quantum field theories provides a rich and interesting playground to study non-local effects in quantum field theory. In particular, in the context of the AdS/CFT correspondence, the $T\overline{T}$ deformation provides a way of moving the CFT into the bulk and thus getting rid of the asymptotically AdS region of the bulk spacetime. Much of the work on this subject so far has focused on the deformed energy spectrum and the partition function. In this paper, we studied the flow of energy eigenstates under the $T\overline{T}$ deformation, and its consequences for the flow of operators, correlation functions, the S-matrix etc. Our results also have a natural 3d gravitational interpretation, which seems closely related to the tensor network approach in AdS/CFT. We will now end with some remarks on potential future directions.

**Entanglement Entropy**

One of the most interesting observables to consider in $T\overline{T}$ deformed theories is the entanglement entropy of a spatial region [21–25]. In ordinary QFT this is already hard to compute and one has to resort to various techniques like the replica trick to do the calculation. In $T\overline{T}$ deformed theories, it is even harder, because these theories are non-local and so many of the techniques useful in the ordinary QFT case may not carry over trivially. Nevertheless, on the plane we can make some more progress now. We have seen that there is a conserved symmetric two-tensor $\widetilde{T}^{(\lambda)}_{\mu\nu}$ that generates all the symmetries that were present in the undeformed theory. In particular, on the plane, the global conformal group is still preserved in the deformed theory. So let us assume that our seed theory is a CFT with central charge $c$, then the modular Hamiltonian associated to a region of size $l$ is given by the spatial integral of the boost operator. In the deformed theory, since there is again global conformal symmetry, it is then tempting to propose that the deformed modular Hamiltonian of an interval of size $l$ for the vacuum state on the plane is given by:

$$\widetilde{K}^{(\lambda)}_l = \int_0^l dx\, \frac{l^2 - x^2}{2l} \widetilde{T}^{(\lambda)}_{00}(0, x) + c_0(\lambda).$$ (5.1)

It seems reasonable that this is the modular Hamiltonian of the reduced state w.r.t. the algebra of dressed operators. Of course, the entanglement entropy of the vacuum is hidden in $c_0(\lambda)$ and it would require a more detailed study to try to extract it. Nevertheless, modular flow with respect to (5.1) is insensitive to $c_0(\lambda)$. This would imply that the modular flow of dressed operators remains unchanged whereas that of the undeformed operators would be highly non-trivial. It would be interesting to study modular flow in these theories in more detail and see how far we can push our techniques to extract $c_0(\lambda)$. We hope to come back to this in future work.

On the cylinder, the flow of operators is much more non-trivial and in particular energy levels can complexify. The question of the computation of entanglement entropy thus becomes

much harder to answer. For instance, when considering the twist operator correlators in the replica trick approach, it is unclear what these twist operators are and whether they are dressed just as any other operator. We suspect this not to be the case, since these twist operators do know about the stress tensor of the theory. From a modular Hamiltonian point of view, it is also complicated, not only because in the undeformed theory we have no expression for the modular Hamiltonian associated to an generic interval, but also since one would again want to extract $c_0(\lambda)$.

**Higher dimensions & other deformations**

Besides the $T\overline{T}$ deformation, there have been various proposals for other *solvable* deformations. For instance, we can apply our formalism to the higher spin generalisations discussed in [4], the $J\bar{T}$ and $T\bar{J}$ deformations considered in [69–73]. For now let us briefly consider the simplest deformation of this kind, namely the marginal $J\bar{J}$ deformation. It is easily seen to be the case that we can write an analogue for $\mathcal{X}$,

$$\mathcal{X}_{J\bar{J}} = \int dy_1 dw_1 G(y_1 - w_1) c_{IJ} J_0^I(y_1) J_0^J(w_1), \tag{5.2}$$

where $I, J$ are flavour indices, with an analogous piece corresponding to $\mathcal{Y}$ in case of the cylinder, which is proportional to the product of the spatial integrals of $J_0$ and $J_1$. It appears that $\mathcal{X}_{J\bar{J}}$ becomes local if $c_{IJ}$ is symmetric. It would naturally also be interesting to apply the techniques in this paper to the single trace version of $T\overline{T}$ [74], which, on the worldsheet, is just a marginal current-current deformation.

Another interesting generalisation is higher dimensions, where an analogous operator to $T\overline{T}$ can be written down. This operator was motivated from holography and has, at least at large $N$, a dual interpretation as moving the boundary inwards. Nevertheless, the factorisation property present in $2d$ only holds at large $N$ in higher dimensions and so it is unclear whether a similar story as presented here holds. Having said that, although the simply rewrite of the spatial integral of the deformation might not be available, the form of the unitary operator in terms of a $d + 1$ dimensional gravity path integral in anti-de Sitter would still exist and it would be interesting to investigate this further, especially with an eye towards holography.

Finally, let us mention the deformation proposed in [11], dubbed $\Lambda_2$-deformation. This deformation is the same as the $T\overline{T}$ deformation, but alongside with it one also turns on a cosmological constant proportional to $1/\lambda^2$ at each step. This feeds non-trivially into the flow of the energy levels. From the Hamiltonian point of view, i.e. we can take the flow of the Hamiltonian to be

$$\partial_\lambda H = \int dy_1 \mathcal{O}_{T\overline{T}}(y_0, y_1) + \frac{(\alpha - 1)L}{8\lambda^2}, \tag{5.3}$$

with $\alpha$ a constant; notice that here we have focused on the cylinder, since on the plane this flow is not well-defined. For this deformation, many of the statements we made in the bulk of the paper still hold. We can still write the analogue of $\mathcal{X}$ and $\mathcal{Y}$. In fact, it is only $\mathcal{Y}$ that changes,

$$\mathcal{Y}_{\Lambda_2} = \mathcal{Y}_{T\overline{T}} + \frac{(\alpha - 1)L}{8\lambda^2}. \tag{5.4}$$

As a consequence, for correlators of dressed operators discussed in 3.2 we can again write down an integral transform for the deformed correlators, which simply takes the undeformed to deformed energy levels. Furthermore, the differential equation for these correlators would be the same as in (3.55), but with an additional $-(\alpha - 1)tL^2\widetilde{G}_\lambda/(4\lambda)^2$ on the right hand side.

**Virasoro symmetry & the theory on the cylinder**

In section 3 we saw that the deformed theory on the plane still enjoys a conformal symmetry whenever the undeformed theory was a CFT. One can wonder whether this lifts to a full Virasoro symmetry. One way to go about this is to analytically continue the deformed theory to Euclidean signature, do radial quantisation and conformally map the plane to the cylinder. One could then define the modes of the stress-tensor and see if they obey the Virasoro algebra.[12] There are two immediate issues with this. First, the analytic continuation is non-trivial, since the theory under consideration is non-local. However, it is plausible that in terms of the dressed operators, the deformed theory can still be regarded as a local theory and such an analytic continuation would work. Second, the conformal map from the plane to the cylinder will introduce a non-trivial space-time dependence in the deformation parameter. This makes the theory on the cylinder (now that we have defined it through the theory on the plane) a $T\overline{T}$ deformed theory with a space-time dependent deformation parameter. This is of course not an issue, but if one wants to define the theory on the cylinder with a space-time-independent coupling $\lambda$, one would have to find a way of getting rid of the space-time dependence of $\lambda$.[13]

On the other hand, the way we have defined the theory on the cylinder in this paper is just the flow of the conserved charges $H$ and $P$. With that definition it seems highly non-trivial to have a Virasoro symmetry. Interestingly, in [10] it was found, through the use of holography at finite cutoff, that there does exist a Virasoro symmetry, albeit a state dependent one. It would be very exciting to see how that Virasoro symmetry emerges in our context.

### Acknowledgement

We thank Edgar Shaghoulian for initial collaboration on this project and discussions. We also thank Alexandre Belin, Jan de Boer, Pawel Caputa, Shouvik Datta, Raghu Mahajan, Gábor Sárosi, Eva Silverstein, Ronak Soni, Herman Verlinde, Sungyeon Yang and Bernardo Zan for discussions and comments on a previous version of the draft. JK is supported by the Simons Foundation.

## A  Lagrangian vs. Hamiltonian definitions

In this appendix, we want to give a classical argument that if we deform the Hamiltonian density infinitesimally, as

$$\mathcal{H}(\phi, \pi) = \mathcal{H}_0(\phi, \pi) + \varepsilon \partial_\lambda \mathcal{H}(\phi, \pi), \tag{A.1}$$

then to leading order in $\epsilon$, this is equivalent to deforming the Lagrangian density of the theory as

$$\mathcal{L}(\phi, \dot{\phi}) = \mathcal{L}_0(\phi, \dot{\phi}) - \epsilon \partial_\lambda \mathcal{H}(\phi, f_0(\dot{\phi})), \tag{A.2}$$

where $\pi = f_0(\dot{\phi})$ is the relation between $\pi$ and $\dot{\phi}$ at $\epsilon = 0$. Note that from the Hamiltonian perspective, we are changing the Hamiltonian density but keeping the symplectic structure of the theory fixed. Consequently, in the Lagrangian perspective, the meaning of the field $\dot{\phi}$ in terms of $\phi$ and $\pi$ changes, but nevertheless the claim is that the Lagrangian density has a simple transformation, as given in (A.2). To show this, we first write

$$\mathcal{L}(\phi, \pi) = \pi \dot{\phi} - \mathcal{H}_0(\phi, \pi) - \epsilon \partial_\lambda \mathcal{H}(\phi, \pi). \tag{A.3}$$

---

[12]Alternatively, one can embed the plane into a diamond on the cylinder and pullback the usual Virasoro modes to that diamond. We thank Jan de Boer for suggesting this.

[13]We thank Edgar Shaghoulian for discussions on this point.

Now we are instructed to solve the EOM of $\pi$ to obtain $\pi$ as a function of $\phi$ and $\dot{\phi}$. Let us assume that this solution takes the form:

$$\pi \equiv f(\phi, \dot{\phi}) = f_0(\phi, \dot{\phi}) + \epsilon f_1(\phi, \dot{\phi}) + O(\epsilon^2). \tag{A.4}$$

By definition, this solves $\dot{\phi} = \frac{\delta \mathcal{H}}{\delta \pi}$, which we can expand perturbatively in $\epsilon$:

$$
\begin{aligned}
\dot{\phi} &= \frac{\delta \mathcal{H}_0}{\delta \pi}(\phi, f_0 + \epsilon f_1) + \epsilon \frac{\delta \partial_\lambda \mathcal{H}}{\delta \pi}(\phi, f_0 + \epsilon f_1) \\
&= \frac{\delta \mathcal{H}_0}{\delta \pi}(\phi, f_0) + \epsilon f_1 \frac{\delta^2 \mathcal{H}_0}{\delta \pi^2}(\phi, f_0) + \epsilon \frac{\delta \partial_\lambda \mathcal{H}}{\delta \pi}(\phi, f_0) + O(\epsilon^2).
\end{aligned} \tag{A.5}
$$

Comparing both sides order by order in $\epsilon$, we learn that

$$\dot{\phi} = \frac{\delta \mathcal{H}_0}{\delta \pi}(\phi, f_0), \quad f_1 \frac{\delta^2 \mathcal{H}_0}{\delta \pi^2}(\phi, f_0) + \frac{\delta \partial_\lambda \mathcal{H}}{\delta \pi}(\phi, f_0) = 0. \tag{A.6}$$

The first equation allows us to determine what $f_0(\phi, \dot{\phi})$ is, the second one then determines $f_1$. So now we have solved for $\pi$ as a function of $\phi$ and $\dot{\phi}$, at least to the leading order in $\epsilon$. We now plug this back into the Lagrangian density:

$$
\begin{aligned}
\mathcal{L} &= \pi \dot{\phi} - \mathcal{H}_0(\phi, \pi) - \epsilon \partial_\lambda \mathcal{H}(\phi, \pi) \\
&= (f_0 + \epsilon f_1) \dot{\phi} - \mathcal{H}_0(\phi, f_0 + \epsilon f_1) - \epsilon \partial_\lambda \mathcal{H}(\phi, f_0 + \epsilon f_1) \\
&= f_0 \dot{\phi} - \mathcal{H}_0(\phi, f_0) + \epsilon f_1 \left( \dot{\phi} - \frac{\delta H}{\delta \pi}(\phi, f_0) \right) - \epsilon \partial_\lambda \mathcal{H}(\phi, f_0) \\
&= \mathcal{L}_0 - \epsilon \partial_\lambda \mathcal{H}(\phi, f_0),
\end{aligned} \tag{A.7}
$$

where in the last line we have used the first EOM in (A.6) to drop the term proportional to $\epsilon f_1$. Therefore, to leading order in $\epsilon$, it is clear that deforming the Hamiltonian is the same thing as deforming the action. The fact that the relation between $\pi$ and $\dot{\phi}$ changes is irrelevant at this order. However, the above argument is entirely classical; perhaps this is sufficient in some large-$N$/semi-classical limit. But in the full quantum theory, we would need to make an argument at the level of the Feynman path integral, and in particular worry about operator ordering ambiguities.

## B  Pressure term in the Energy flow

Here we wish to check that

$$\langle n | T_{xx} | n \rangle = -\partial_L E_n. \tag{B.1}$$

This is a crucial input in Zamolodchikov's argument [1] for the flow of energy eigenvalues. In order to prove this, let us begin by computing the following torus one point function:

$$f(\beta, L) \equiv \langle T_{xx} \rangle_{T^2} = \sum_n d_n e^{-\beta E_n(L)}{}_L \langle n | T_{xx} | n \rangle_L, \tag{B.2}$$

where $\beta$ is the temperature, $L$ is the length of the spatial circle, and $d_n$ is the degeneracy of the $n$th energy level, which we will assume is $L$-independent. The subscript $L$ on the eigenstates denotes the length of the circle on which the system lives. Assuming local rotation invariance, we can also view this one point function by turning it on its side, i.e., interpret the $x$ direction

as Euclidean time and the $\tau$ direction as space. From this perspective, we are evaluating the one-point function of the energy density:

$$f(\beta, L) = \sum_m \frac{d_m}{\beta} E_m(\beta) e^{-LE_m(\beta)}, \tag{B.3}$$

where note that $E_m(\beta)$ is the energy on a circle of radius $\beta$. Comparing these two expressions, we arrive at

$$\sum_n d_n e^{-\beta E_n(L)} \langle n|T_{xx}|n\rangle_L = \sum_m \frac{d_m}{\beta} E_m(\beta) e^{-LE_m(\beta)}. \tag{B.4}$$

We now multiply both sides by $e^{\beta E_p(L)}$ and integrate $\beta$ along the imaginary axis. On the left hand side, this picks out the contribution of the $p$th energy level. If we further *assume* that the stress tensor one point function is the same within all the degenerate states at that energy, then we get,

$$
\begin{aligned}
d_p \delta(0)\, _L\langle p|T_{xx}|p\rangle_L &= -i \int_{-i\infty}^{i\infty} d\beta \sum_m \frac{d_m}{\beta} E_m(\beta) e^{\beta E_p(L) - LE_m(\beta)} \\
&= -i \int_{-i\infty}^{i\infty} d\beta \frac{1}{\beta} \left(-\partial_L Z(L,\beta)\right) e^{\beta E_p(L)} \\
&= i\partial_L \int_{-i\infty}^{i\infty} \frac{d\beta}{\beta} Z(L,\beta) e^{\beta E_p(L)} - i(\partial_L E_p) \int_{-i\infty}^{i\infty} d\beta Z(L,\beta) e^{\beta E_p(L)}.
\end{aligned} \tag{B.5}
$$

The second term above is proportional to the inverse Laplace transform, and in the present case simply gives $d_p\,\delta(0)$. In the first term, we need to confront the following integral:

$$
\begin{aligned}
\text{1st term} &= \partial_L \int_{-i\infty}^{i\infty} \frac{d\beta}{\beta} \sum_m d_m e^{\beta(E_p(L) - E_m(L))} \\
&\sim \partial_L \sum_m d_m \Theta(E_p(L) - E_m(L)) \\
&= \partial_L \sum_{m \leq p} d_m. 
\end{aligned} \tag{B.6}
$$

This is simply the number of states below the energy level $E_p$. Since these degeneracies are $L$-independent, the $\partial_L$ outfront kills this term, and so we obtain the desired formula. This argument assumes only translation plus rotation invariance, and that the stress tensor one point function is independent of any internal degeneracy of energy eigenstates.

## C  Fixing the stress-tensor flow

In this appendix we give some more details on the flow of the stress-tensor for the case of a spatial slice being a circle of length $L$. The equation we want to solve is

$$\partial^\mu D_\lambda \widetilde{T}^{(\lambda)}_{\mu\nu}(x) = i[\mathcal{Y}(x_0), \widetilde{T}^{(\lambda)}_{0\nu}(x)]. \tag{C.1}$$

The general solution is given by

$$D_\lambda \widetilde{T}^{(\lambda)}_{\mu\nu} = \mathcal{F}_{\mu\nu} + \mathcal{A}_{\mu\nu}, \tag{C.2}$$

with $\partial^\mu \mathcal{F}_{\mu\nu} = i[\mathcal{Y}(x_0), \widetilde{T}^{(\lambda)}_{0\nu}(x)]$ and $\partial^\mu \mathcal{A}_{\mu\nu} = 0$. We can write the commutator on the right-hand side of (C.1) as a sum of a spatial derivative and a temporal derivative (by introducing an integral from $-\infty$ to $x_0$). Equating these derivatives with those on the left-hand side of (C.1), we find

$$\mathcal{F}_{0\nu} = -\frac{i}{L} \int_{M_{x_0}} d^2y \, [\{H, T_{11}(y_0, y_1)\}, T_{0\nu}(y_0, x_1)], \quad \mathcal{F}_{1\nu} = -\frac{2}{L}\{P, T_{0\nu}(x)\}, \qquad \text{(C.3)}$$

with $M_{x_0} = (-\infty, x_0] \times S^1$. It remains to find an appropriate $\mathcal{A}_{\mu\nu}$. It is convenient to directly solve the divergenceless condition for $\mathcal{A}$. Let us therefore write $\mathcal{A}_{0\nu} = A_\nu$ and $\mathcal{A}_{1\nu} = B_\nu$, so that

$$A_\nu(x_0, x_1) = -\int_{-\infty}^{x_0} dx_0' \partial_1 B_\nu(x_0', x_1) + \phi_\nu, \qquad \text{(C.4)}$$

where we included a constant piece $\phi_\nu$. We now fix $\phi_\nu$ and $B_\nu$ by requiring consistency with $\partial_\lambda H = \int dy_1 \mathcal{O}_{T\bar{T}}$, $\partial_\lambda P = 0$, and symmetricity in the $\mu\nu$ indices. Notice that the second condition is equivalent to covariant constancy of $P$. As we have done in the main text, we will assume that $H$ and $P$ are the generators of temporal and spatial translations, respectively. We get the following conditions on $\phi_\nu$ and $B_\nu$

$$\phi_0 = -2\left(\frac{P}{L}\right)^2, \quad \phi_1 = 0 \qquad \text{(C.5)}$$

$$B_0(x) = \frac{2}{L}\{\widetilde{T}^{(\lambda)}_{00}(x), P\}, \quad \partial_1 B_1(x) = \frac{i}{L}[\{H, \int dy_1 \widetilde{T}^{(\lambda)}_{11}(x_0, y_1)\}, \widetilde{T}^{(\lambda)}_{01}(x)]. \qquad \text{(C.6)}$$

This makes the flow for the stress-tensor rather complicated, especially the flow for $T_{11}$. However, a unique smooth solution always exists by matching on the undeformed theory at $\lambda = 0$ and noticing that the spatial integral of the right-hand-side of the second equation in (C.6) vanishes and so there is no zero-mode issue due to the compactness of the spatial slice. The solution can thus obtained by inverting $\partial_1$ by using the Green function on a circle of length $L$,

$$B_1(x) = \frac{i}{L}\left[\{H, \int dy_1 \widetilde{T}^{(\lambda)}_{11}(x_0, y_1)\}, \int_0^L dw_1 G(x_1 - w_1) \widetilde{T}^{(\lambda)}_{01}(x_0, w_1)\right], \qquad \text{(C.7)}$$

where $G(x_1 - y_1) = \frac{1}{2}\text{sgn}(x_1 - y_1) - (x_1 - y_1)/L$. The final flow of the stress tensor on the cylinder is thus given by

$$D_\lambda \widetilde{T}^{(\lambda)}_{00}(x) = -\frac{i}{L}\int_{M_{x_0}} d^2y \, [\{H, \widetilde{T}^{(\lambda)}_{11}(y_0, y_1)\}, \widetilde{T}^{(\lambda)}_{00}(y_0, x_1)] - Q(x) \qquad \text{(C.8)}$$

$$D_\lambda \widetilde{T}^{(\lambda)}_{01}(x) = D_\lambda \widetilde{T}^{(\lambda)}_{10}(x) = 0 \qquad \text{(C.9)}$$

$$D_\lambda \widetilde{T}^{(\lambda)}_{11}(x) = -\frac{2}{L}\{P, \widetilde{T}^{(\lambda)}_{01}(x)\} + B_1(x), \qquad \text{(C.10)}$$

with

$$Q(x) = 2\left(\frac{P}{L}\right)^2 + \int_{-\infty}^{x_0} dx_0' \partial_1 B_0(x_0', x_1). \qquad \text{(C.11)}$$

# D   Details on $U$ as $3d$ path integral

Using the foliation in 4.7, we can write 4.6 in terms of geometric data. The extrinsic curvature of the constant $\lambda$ hypersurfaces are,

$$K_{ab} = \frac{1}{\lambda}(\gamma_{ab} - \lambda\partial_\lambda\gamma_{ab}), \quad K^{ab} = \lambda(\gamma^{ab} + \lambda\partial_\lambda\gamma^{ab}), \quad K = g^{ab}K_{ab} = (2 - \lambda\gamma^{ab}\partial_\lambda\gamma_{ab}) \text{ (D.1)}$$

and so

$$K_{ab}K^{ab} - K^2 = (-2 - \lambda^2\partial_\lambda\gamma_{ab}\partial_\lambda\gamma^{ab} + 2\lambda\gamma^{ab}\partial_\lambda\gamma_{ab} - \lambda^2(\gamma^{ab}\partial_\lambda\gamma_{ab})^2). \tag{D.2}$$

Which allows us to rewrite the integrand in 4.6 as

$$-\partial_\lambda\gamma_{ab}\partial_\lambda\gamma^{ab} - (\gamma^{ab}\partial_\lambda\gamma_{ab})^2 = \frac{1}{\lambda^2}(K_{ab}K^{ab} - K^2) + \frac{2}{\lambda^2} - \frac{2}{\lambda}\gamma^{ab}\partial_\lambda\gamma_{ab}. \tag{D.3}$$

And, since $\sqrt{g} = \frac{1}{2\lambda^2}\sqrt{\gamma}$ we have

$$S = -\frac{1}{16}\int d^3x\sqrt{g}\left[2(K_{ab}K^{ab} - K^2) + 4 - 4\lambda\gamma^{ab}\partial_\lambda\gamma_{ab}\right]. \tag{D.4}$$

Furthermore,

$$\int d^3x\sqrt{g}\lambda\gamma^{ab}\partial_\lambda\gamma_{ab} = \int d^3x\frac{1}{\lambda}\partial_\lambda\sqrt{\gamma} = \int d^2x\left.\sqrt{g_0}\right|_{\lambda=0}^{\lambda} + 2\int d^3x\sqrt{g}, \tag{D.5}$$

with $g_0$ the induced metric on a constant $\lambda$ slice. Moreover, the Gauss-Codazzi equations tell us,

$$K_{ab}K^{ab} - K^2 = -R^{(3)} + R^{(2)} + 2\nabla_c(\nabla_d n^c n^d - n^c\nabla_d n^d), \tag{D.6}$$

with $R^{(3)}$ the three dimensional curvature and $R^{(2)}$ the two dimensional one. Plugging this into D.4 and using $R^{(2)} = \lambda R^{(\gamma)}$, we get

$$2\int d^3x\sqrt{g}(K_{ab}K^{ab} - K^2) = -2\int d^3x\sqrt{g}R^{(3)} + 2\int d^3x\sqrt{\gamma}R^{(\gamma)}$$
$$+ 4\int_\partial d^2x\sqrt{g_0}n_c(\nabla_d n^c n^d - n^c\nabla_d n^d). \tag{D.7}$$

The first term of the boundary term is zero because the norm of $n_c$ is constant and the second term is $K = \nabla_\mu n^\mu$ for the boundary at $\lambda = 0$ and with the opposite orientation for the boundary at $\lambda = \lambda_c$, so that the normals are always inwards (inwards in the annulus) pointing. Plugging this in (D.4) we find the promised result,

$$S = \frac{1}{16\pi G_N}\left[\int d^3x\sqrt{g}(R + 2) + 2\int_\Sigma\sqrt{g_0}d^2x(K - 1)\right.$$
$$\left. - 2\int_{\hat{\Sigma}}d^2x\sqrt{\hat{g}_0}\hat{K} - \int d^3x\sqrt{\gamma}R^{(\gamma)}\right]. \tag{D.8}$$

Here $\Sigma$ is the timelike boundary at $\lambda = 0$ and some finite $\lambda$, $\hat{\Sigma}$ the spacelike boundary at $t = 0$, the other hatted quantities denoting the corresponding objects on $\hat{\Sigma}$ and $R^{(\gamma)}$ the curvature of $\gamma$.

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
