# Peer review of "On the flow of states under T¯T"

_SciPost Physics, doi:SciPost Phys. 9, 078 (2020)_

## Round 1 · Referee Report · John Cardy (Referee 1) · 2020-8-7

Strengths

  1. A significant contribution to the subject
  2. Well written and explained

Weaknesses

  1. somewhat formal in its approach
  2. fails to discuss UV divergences
  3. one result possibly empty of content

Report

  1. This paper is an interesting addition to the literature on TTbar. It adopts a novel approach in using the operator-state description rather than the euclidean path integral, and in focussing on the deformed states rather than the spectrum, although the method does rederive previous results on the latter.

  2. The paper relies heavily on a method introduced in [17] and not surprisingly yields results which are similar, although expressed in the language of states rather than operators of the path integral approach. However eq. (1.2), which is one way of expressing the solvability of the deformation, is very elegant, especially in the way it shows the difference between the full line and the compactified circle, a difference which was not directly addressed in [17].

  3. However I do find it strange that nowhere is there a discussion of potential UV divergences in the somewhat formal expressions which appear. Indeed it was argued extensively in [17] that these do appear when local operators are dressed by a string, and that they may be regulated either by point-splitting or by regulating the Green's function. But as they are written, the formal expressions for the deformed operators simply do not exist.

  4. i also do not understand the second way of defining dressed operators by conjugation with U. This leads to the somewhat paradoxical conclusion that the conformal symmetry of the seed theory is preserved, if hidden. This is reminiscent of the (false) statement that an interacting QFT can be thought of as a dressed version of the free theory and therefore retains all of its algebraic features, which is clearly wrong, largely because of UV divergences which mean that the Hilbert spaces are totally different. It is true that the cylinder spectrum does retain memory of the Virasoro algebra, but that is UV finite unlike local operators. I suspect that the conclusions in this part of the paper are merely formal and possibly erroneous as stated.

  5. The last part of the paper contains interesting ideas about the tensor network interpretation.

Requested changes

  1. The authors should explicitly discuss the matter of UV divergences at appropriate points and how they might modify their results.
  2. They should address point 4 above, which might lead to significant changes in this section.

---

## Round 1 · Referee Report · Anonymous (Referee 2) · 2020-8-23

Strengths

1- Clearly written. 2- The Hamiltonian point of view is less commonly applied to TTbar than the Lagrangian point of view. 3- Several detailed discussions of relations with other work (on the S-matrix, energy levels, 3d gravity) and an especially interesting comparison with tensor networks.

Weaknesses

1- The text does not make it entirely clear which of the results are new compared to [17]. 2- Many expressions are formal and may be ill-defined due to UV divergences. 3- On the plane, dressed operators seem to just be operators of the undeformed theory.

Report

My apologies to the authors for the delay in reviewing their interesting paper.

1- The authors find that the TTbar deformation of 2d field theories on the plane formally amounts to a unitary transformation on the Hilbert space, while it is more elaborate on the cylinder. From this very interesting point of view they rederive various results on energy levels, S-matrix, and correlators. It is not clearly stated which results are new, except for the flow of states themselves.

2- Around (3.26) it is stated that deforming by the dressed $\tilde{T}\tilde{\bar{T}}$ will be equivalent to a one-shot shift by $\lambda T\bar{T}$. This seems very close to the classical claims, where the classical TTbar-deformed action is given in worldsheet coordinates by the one-shot-deformed action: see eq (2.10) of [Coleman, Aguilera-Damia, Freedman, Soni arXiv:1906.05439], building upon work by Tateo et al.

3- More generally, the authors should make an effort to relate their Cauchy string variable X to previous literature, such as Cardy's random metric reformation of TTbar (also using Hubbard-Stratanovich fields), or the JT-gravity approach (or massive gravity, following Tolley).

4- On the plane, it seems dressed operators are simply operators of the original undeformed theory. It is not clear that their correlators teach us anything about the TTbar-deformed theory. If one sees the TTbar theory as viewing the original theory from a worldsheet (à la dynamical change of coordinates of Tateo et al), then the dressed operators are simply forgetting the worldsheet and directly working with the original theory.

5- I share the first reviewer's worries about UV divergences.

6- I must correct the first referee in that treating the TTbar deformation in the Hamiltonian formalism is not novel, see for instance [Le Floch, Mezei arXiv:1907.02516] which heavily relies on it. However, it is true that tracking the state quite so precisely as the present submitted paper is novel.

7- Section 4.2 ends without really giving a holographic interpretation: the authors make the Einstein-Hilbert action appear, but there remains a rather under-explained path-ordered exponential. Besides, (I'm probably just being confused here), isn't the Einstein-Hilbert action only the first three integrals in (4.8)? I don't understand how to interpret the $R^{(\gamma)}$ term geometrically.

8- The authors provide a very suggestive comparison with tensor networks that seems promising, but I am not knowledgeable about that aspect.

Requested changes

1- Compare their X with the literature describing the classical TTbar deformation as a dynamical change of coordinates.

2- Relatedly, clarify whether their dressed operators are non-trivial (on the plane, say) or are simply the original operators in the original theory.

3- Discuss UV divergences.

4- Correct the following minor points

4a- Using sgn(x) in eq (2.6) is weird: in the interval (0,L) mentioned below the equation the sign is just +1, and sgn(x) is not enough to deal with arbitrary real x. One should either just write -x/L for x in (0,L), or write something with the fractional part of x to make it valid for all real x and L-periodic.

4b- Second line of (2.8) should have the opposite sign.

4c- Below (2.22), replace reference to eq (2.22) by (2.21). Inside the integrals in (2.24) and (2.33), replace lambda by lambda' (twice in each). Missing exp(epsilon s) next to O TTbar in (2.27). Between (2.27) and (2.28), replace reference to eq (2.26) by (2.11).

4d- In (2.29) the first $\xi$ and $Q$ should either both or neither have the superscript $a$; the first $S$ is missing $\phi$ as an argument. In (2.30) change $(Q^1)_k$ to $(Q^1_k)$. The $(Q^1)^2$ and $Q^1P$ terms seem to give $P^2$ without factor of~$2$ once integrated out. Below (2.30) it should be stated explicitly that $\xi$ has vanishing spatial integral.

4e- In (3.9), $x-y$ should be $x_1-y_1$.

4f- In section 4.2 missing parentheses around 2.25, 2.3.

4g- Typos: degenracy, cirlce, ``Let the seed theory by'', detials

---

## Round 2 · Referee Report · John Cardy (Referee 1) · 2020-10-31

Strengths

1. This paper gives a potentially very useful way of understanding how states evolve under TTbar.

Weaknesses

1. See below.

Report

1. If anything, the paper is overlong and contains too much speculative material towards the end which obscures the main result and could have been deferred until the authors have more definitive results. However, this should not disqualify the paper from being published.

Requested changes

1. None

---

## Round 2 · Referee Report · Anonymous (Referee 3) · 2020-11-5

Strengths

1- New result on evolution of states.
2- Clear and detailed.
3- Somewhat more explicit discussion (than in v1) of relation between their dressed operators and the holographic interpretation.

Weaknesses

1- Some of the later speculations seem like they could be postponed to another paper (however I understand why the authors may want to get them published now).

Report

As far as I can tell, the revision addresses adequately both my comments and Professor Cardy's comments.

---

## Round 2 · Author Response

Dear Editor,

We thank the referees for their careful reading of our manuscript and their valuable comments. The manuscript is modified quite a bit to address the concerns expressed in both the reports, although the main results have remained unaffected. See below for our reply to both reports.

Report 1

We thank Prof. Cardy for his valuable questions and comments.

1) The first clarification the referee suggested was about the analysis of the ultraviolet (UV) divergences. We agree that our original manuscript lacked a careful discussion on this issue and we deeply appreciate this comment made by the referee. In order to address it we have, at various places in the revised manuscript, discussed the UV divergences. First, when we introduce the Green function G in equation 2.4 and below, we mention that the coincident point limit needs to be regulated in order for equation 2.7 to make sense. We mentioned this below 2.7. The next place where UV divergences play a role is around equation 3.8. Since our flow equation for the undeformed operators agrees with that of Prof. Cardy's from [17], we make use of his analysis of the UV divergences at this point to address the need for operator renormalization. Similarly, we also discuss the dressed operators and their divergencies below equation 3.17.

2) The second clarification sought by the referee is about the definition of dressed operators. Here we mention that we are not saying that the T¯T deformed theory can always be written as a dressed version of the undeformed theory. Let us first consider the classical case for simplicity. In the classical theory on the plane (and only on the plane) and within the finite energy sector, it so happens that the deformed Hamiltonian can be obtained from a canonical transformation of the undeformed Hamiltonian, see eq. 3.11. Classically, this means that the deformation is merely implementing a canonical transformation on the phase space coordinates. It is this fact, which allows us to define the dressed operators -- they are the observables written in the canonically transformed coordinates. Already at this stage, we point out that this is \emph{not} the same as a statement along the lines suggested by the referee, i.e., 'interacting QFT is a dressed version of a free theory''. In that case, the interacting Hamiltonian is not a canonical transformation of the free Hamiltonian. Quantum mechanically, the deformation implements a Bogoliubov transformation, but we can nevertheless define dressed observables. These dressed operators do indeed have UV divergences in their flow equation -- we expect these divergences to be related to operator renormalization. Importantly, the correlation functions of these operators are finite. We have tried to reiterate some of the points above in the draft, for instance below equation 3.17.

Let us emphasize that on the cylinder this statement about the canonical/Bogoliubov transformation is not true and we have no precise claim to make. We added a comment about this below 3.26.

Report 2

We thank the referee for her/his valuable questions and comments.

1) We have done so below 4.6 in the revised manuscript 2) Here, the referee sought a clarification for whether the dressed operators are non-trivial. The dressed operators are not the same as the original, undeformed CFT operators. Indeed, these latter ones are discussed in the subsection titled Undeformed operators'' on p 15. Consider first the classical theory. It makes sense to define dressed observables only because the deformed Hamiltonian is a canonical transformation of the original Hamiltonian; see equation 3.11. The canonical transformation (generated by X) defines a new set of phase space coordinates at every point along the flow -- these are the dressed observables. Note that this definition only makes sense within the finite energy sector on the plane . On the contrary, for the cylinder, or even for finite energy density states on the plane, the T¯T deformation is not a pure canonical transformation as we cannot drop the Y term in this case, and so it is not clear how to define dressed observables. 3) We have discussed the issue of UV divergences in the revised manuscipt, see our response to report number one (point number one thereof). 4) Thank you for spotting all the typos, we have corrected all of them in the revised manuscript.

---

## Editorial Decision

published